# QTIP: Quantization with Trellises and Incoherence Processing

**Albert Tseng**
Cornell University
albert@cs.cornell.edu

**Qingyao Sun**
Cornell University
qs234@cornell.edu

**David Hou**
dhou@alumni.caltech.edu

**Christopher De Sa**
Cornell University
cdesa@cs.cornell.edu

## Abstract

Post-training quantization (PTQ) reduces the memory footprint of LLMs by quantizing weights to low-precision datatypes. Since LLM inference is usually memory-bound, PTQ methods can improve inference throughput. Recent state-of-the-art PTQ approaches use vector quantization (VQ) to quantize multiple weights at once, which improves information utilization through better shaping. However, VQ requires a codebook with size exponential in the dimension. This limits current VQ-based PTQ works to low VQ dimensions ($\leq 8$) that in turn limit quantization quality. Here, we introduce QTIP, which instead uses trellis coded quantization (TCQ) to achieve ultra-high-dimensional quantization. TCQ uses a stateful decoder that separates the codebook size from the bitrate and effective dimension. QTIP introduces a spectrum of lookup-only to computed lookup-free trellis codes designed for a hardware-efficient "bitshift" trellis structure; these codes achieve state-of-the-art results in both quantization quality and inference speed.

## 1 Introduction

Large language models (LLMs) have accelerated advancements in fields ranging from natural language processing [34] to scientific modeling [28]. However, the largest LLMs have hundreds of billions of parameters that can take over a terabyte of memory to load in half-precision; this size poses significant challenges for the practical deployment of LLMs [33, 17, 2]. For example, small-batch autoregressive decoding, a common form of inference for LLMs, is memory bound [35]. Even on a modern datacenter GPU with $\approx$ 3TB/s memory bandwidth, a large LLM ($>$ 200GB) can only be directly run at $<$ 20 tokens per second and may require multiple devices [35]. One way to accelerate inference is by compressing LLMs. This directly reduces the memory footprint of the model and increases the theoretical maximum inference throughput on any given machine.

One form of compression, weight-only post-training quantization (PTQ), quantizes trained model weights to lower precision datatypes [9, 35, 5]. The latest state-of-the-art weight-only PTQ methods, QuIP# and AQLM, use vector quantization (VQ) to achieve high-quality 2-bit models [35, 11]. In VQ, a vector $x \in \mathbb{R}^d$ is quantized to one of $2^{kd}$ vectors in $\mathbb{R}^d$ that form a codebook $C \in \mathbb{R}^{2^{kd} \times d}$. A higher vector dimension $d$ allows for better codebook shaping and packing density, improving information utilization [19]. However, unstructured VQ requires exponential time and space in both the bitrate and dimension, limiting its practicality. During quantization, VQ costs $O(2^{kd}d)$ time to perform nearest-neighbor rounding to $C$, and during inference, $C$ must fit in hardware cache for fast lookups. This exponential scaling limits how high $d$ can be and thus the advantages of VQ over scalar quantization.

38th Conference on Neural Information Processing Systems (NeurIPS 2024).

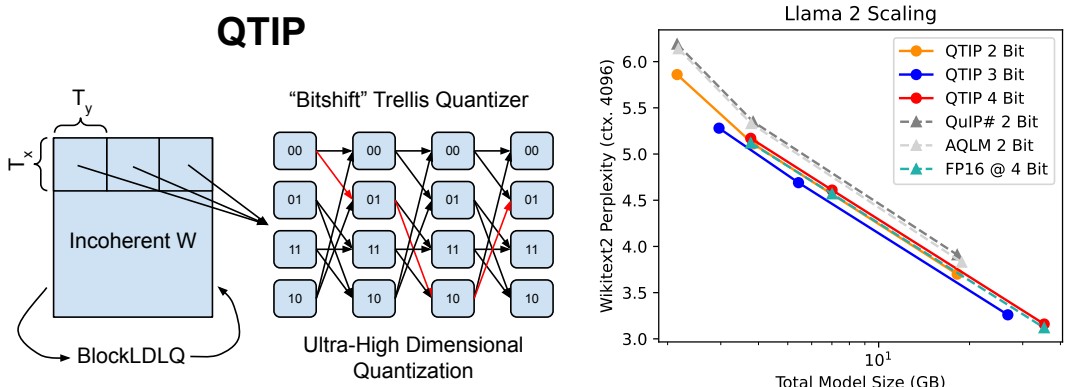

Figure 1: QTIP performs ultra-high dimensional ($> 100$) quantization by using Trellis Coded Quantization, which has linear cost in dimension. This enables QTIP to outperform Vector Quantization-based approaches (QuIP#, AQLM) that are limited to low dimensions. With QTIP, 2 bit models scale better than theoretically optimal 4 bit models.

To address this limitation, we propose QTIP, which uses trellis-coded quantization (TCQ) to enable tractable ultra-high-dimensional ($> 100$) quantization and improve quantization quality over prior VQ-based approaches. In the simplest scalar form of TCQ, a length-$T$ sequence $S$ is *statefully* quantized using a trellis – a directed graph $G$ with $2^L$ nodes, each with $2^k$ incoming and outgoing edges and a scalar value [24]. The reconstructed sequence $\hat{S}$ corresponds to the node values of a length-$T$ walk on $G$, and quantization finds the walk that minimizes some distortion metric on $S$ and $\hat{S}$. Since neighboring entries in $\hat{S}$ are connected by one of $2^k$ edges, we only need to store *which edge* an entry came from, which takes $k$ bits. For additive distortion metrics such as squared error, the optimal $\hat{S}$ can be found with the Viterbi algorithm, which runs in $O(2^L T)$ time [13, 24]. This means that the cost of quantization is *independent* of the bitrate $k$ and *linear* in the sequence dimension $T$, enabling tractable high dimensional quantization.

However, TCQ is not free. During inference, vanilla TCQ requires storing both $G$ and the size $2^L \times V$ node value codebook, which can be too large to fit in cache. TCQ-quantized sequences also cannot generally be decoded in parallel, as $t$th elment of $\hat{S}$ could depend on up to the first $tk$ encoded bits. In QTIP, we solve these issues by introducing a series of fast compute-based Gaussian codes designed for the hardware-efficient "bitshift trellis." Specifically, the bitshift trellis supports parallel decoding, does not require storing $G$, and our compute-based codes eliminate needing to store a large node value codebook. This enables high-quality quantization of Gaussian sources while supporting fast inference, and we adopt incoherence processing with the random Hadamard transform to ensure that LLM weights are approximately i.i.d Gaussian distributed. Altogether, QTIP

1. Achieves a state-of-the-art combination of weight-only LLM PTQ quality and fast inference through hardware-efficient trellis and codebook design.

2. Introduces multiple novel hardware-efficient ($\leq 4$ instructions per weight) compute-based random Gaussian codes for TCQ on i.i.d. Gaussian sources.

## 2 Background and Related Works

We focus on weight-only post-training quantization (PTQ) of LLMs in this work; other model-compression approaches include quantization-aware training (QAT) and pruning. These methods are not strictly orthogonal to each other, as one could both prune and quantize a model. Since QTIP is a weight-only PTQ method, the rest of this section focuses on this area. Most current state-of-the-art PTQ methods round to minimize the per-layer proxy loss from Nagel et al. [27].

$$\ell(\hat{W}) = \mathbb{E}_x \left[ \|(\hat{W} - W)x\|^2 \right] = \text{tr} \left( (\hat{W} - W)H(\hat{W} - W)^T \right) \tag{1}$$

Here, $\hat{W} \in \mathbb{R}^{m \times n}$ is the quantized weight matrix, $x \in \mathbb{R}^n$ is an input activation, and $H = \mathbb{E}_x\left[xx^T\right] \in \mathbb{R}^{n \times n}$ is interpreted as a proxy Hessian matrix. This objective is defined *per-layer*, making it tractable for very large models. However, minimizing it is difficult due to the non-differentiable nature of quantization. Instead many works have proposed algorithms such as Hessian-based adaptive rounding, alternating optimization, and even coordinate descent to approximately minimize the proxy error [11, 5, 35, 14].

## 2.1 Incoherence Processing

The effectiveness of these methods depends on properties of $W$. For example, many works have observed that weight and activation outliers cause poor quantization quality [10, 20, 29]. In QuIP, Chee et al. [5] proposed that *incoherence* was important for quantifying this effect.

**Definition 2.1** (Chee et al. [5]). A Hessian $H \in \mathbb{R}^{n \times n}$ is $\mu$-incoherent if its eigendecomposition $H = Q\Lambda Q^T$ has $\max_{i,j} |Q_{ij}| = \max_{i,j} |e_i^T Q e_j| \leq \mu/\sqrt{n}$. A weight matrix $W \in \mathbb{R}^{m \times n}$ is $\mu$-incoherent if $\max_{i,j} |W_{ij}| = \max_{i,j} |e_i^T W e_j| \leq \mu \|W\|_F/\sqrt{mn}$.

Essentially, incoherence means the weights and important rounding directions (Hessian eigenvectors) are not too large in any direction, aiding quantization. To make $W, H$ incoherent (small $\mu$), one can perform *incoherence processing* (IP) by conjugating $W, H$ with random orthogonal matrices $U, V$: $\tilde{W} \leftarrow UWV^T, \tilde{H} \leftarrow VHV^T$. QuIP# introduced IP with the random Hadamard transformation (RHT), which performs $\tilde{W} \leftarrow V_m S_m W S_n V_n^T, \tilde{H} \leftarrow V_n S_n H S_n V_n^T$ where $V_k$ is a $k \times k$ Hadamard matrix and $S_k$ is a length $k$ random sign vector. The RHT achieves, with probability $\geq 1 - \delta$, $\mu_{\tilde{W}} = 2\log(4mn/\delta)$, meaning that $\tilde{W}$'s entries are approximately independently Gaussian distributed, which can aid quantization [35, 3]. We choose to build on incoherence processing here because the independent Gaussian-like weights it produces are suitable inputs for trellis coding [23].

## 2.2 Vector Quantization (VQ) for LLM PTQ

$k$-bit VQ quantizes a $d$ dimensional vector $S$ to one of $2^{kd}$ $d$-dimensional vectors that form a codebook $C \in \mathbb{R}^{2^{kd} \times d}$ [1]. Since $C$ is an unstructured collection of arbitrary vectors, VQ enables better shaping and packing density than scalar product quantization (SPQ), where each entry in $S$ is quantized independently [19]. However, this also comes at the cost of exponential time quantization and exponential space inference: finding the nearest neighbor in $C$ requires $O(2^{kd}d)$ time, and storing $C$ requires $O(2^{kd}d)$ space. The current crop of state-of-the-art LLM PTQ methods, QuIP# and AQLM, both use VQ to achieve high-quality 2-bit models. Since the shaping advantage of VQ comes from high dimensionality, both QuIP# and AQLM attempt to maximize dimensionality. AQLM's uses a large 8D codebook (1MiB) that does not fit in L1 cache. QuIP# uses an 8D compressible codebook based on the $E_8$ lattice, which is highly symmetric. This codebook is compressible by $256\times$ and barely fits in L1 cache. In either case, the VQ dimension is effectively hardware-limited to $\leq 8$, motivating methods that enable even higher-dimensional quantization.

## 2.3 Trellis-Coded Quantization (TCQ)

TCQ was first proposed by Marcellin and Fischer [24] to apply the benefits of trellis coded *modulation*, a conceptually dual problem, to quantization. Define a $(L, k, V)$ trellis $G$ as a directed graph with $2^L$ nodes, each of which has $2^{kV}$ incoming and outgoing edges and a value $\in \mathbb{R}^V$; these values form a codebook $C \in \mathbb{R}^{2^L \times V}$. To quantize a length-$T$ sequence $S \in \mathbb{R}^T$, each contiguous length-$V$ subsequence of $S$ is assigned to a node $\in G$, with the restriction that the assigned nodes form a walk. The reconstruction $\hat{S}$ of $S$ is then given by concatenating node values in the walk. When $V = 1$, this setup describes Marcellin and Fischer [24]'s original scalar TCQ. When $V > 1$, this describes TCVQ, which applies TCQ to vectors [12, 37].

Finding the optimal $\hat{S}$ under an additive distortion metric can be done with the Viterbi algorithm in $O(2^L T)$ time. This is linear in sequence length, enabling ultra-high dimensional quantization. For exposition, we briefly describe the Viterbi algorithm here. Concretely, if we want to quantize a $T$-length scalar sequence reinterpreted as a sequence of vectors $s_1, s_2, \ldots, s_{T/V} \in \mathbb{R}^V$ using a trellis

code with graph $G$ and codebook $C$, this corresponds to solving the optimization problem

$$\text{minimize} \sum_{i=1}^{T/V} \|C_{x_i} - s_i\|^2 \quad \text{over } x_1, x_2, \dots, x_{T/V} \text{ the vertex sequence of a walk on graph } G.$$

This optimization problem can be solved exactly with dynamic programming via the value function

$$\mathcal{V}_t(x) = \min \left\{ \sum_{i=1}^{t} \|C_{x_i} - s_i\|^2 \,\middle|\, x_1, x_2, \dots, x_t \text{ the vertex sequence of a walk on } G \text{ and } x_t = x \right\}$$

using the update rule

$$\mathcal{V}_t(y) = \min_{(x,y) \in G} \mathcal{V}_{t-1}(x) + \|C_y - s_t\|^2.$$

This Viterbi approach clearly takes time linear in $T$ and in the number of edges of $G$; with a few simple optimizations this can be brought to $O(2^L T)$. In comparison, brute-force-searching all possible $2^{kT}$ codes—which is what we would need to do for an unstructured $k$-bit $T$-dimensional codebook—would take time proportional to $2^{LT/V}$. The ability to tractably find the closest representable vector in $\mathbb{R}^T$, even for large $T$, is in some sense the "main benefit" of trellis coding. For i.i.d sources, as $L$ increases, TCQ efficiently approaches the infinite-length distortion-rate $D_R$, which lower bounds the attainable distortion of a $k$-bit quantizer [19]. As shown in Table 1, when quantizing an i.i.d. Gaussian with $k = 2$, the scalar Lloyd-Max quantizer attains 0.118 MSE, QuIP#'s 8D E8P codebook 0.089 MSE, our (QTIP) 256D $L = 16$ TCQ quantizer 0.069 MSE, and $D_R = 0.063$ [21, 25, 35, 8].

## 3 QTIP

Quantizing with TCQ requires storing both the codebook ($2^L \times V$) and trellis structure ($2^L \times 2^{kV}$) during inference. These components are too large for fast inference when $L \gtrsim 12$, which is necessary for high quality. Furthermore, for a generic trellis, recovering the state (and so the decoded value) at step $t$th requires a graph walk using the first $kt$ bits: this prevents parallel decoding. QTIP solves these problems with a novel combination of incoherence processing, a hardware-efficient "bitshift trellis," and fast compute-based random Gaussian codes. Incoherence processing makes $W$ approximatelly i.i.d Gaussian, which reduces quantization to Gaussian source coding. The bitshift trellis removes needing to store the trellis structure during decoding and also enables parallel decoding. Finally, the fast compute-based random Gaussian codes remove the need to store the full codebook, completing the equation for fast inference. On the quality side, the fast random Gaussian codes enable the simple bitshift trellis to match complicated trellises and achieve state-of-the-art quantization quality.

The main focus of QTIP is on *what to quantize with* (i.e. TCQ) and not *how to quantize* (e.g. adaptive rounding or descent methods). The general construction of QTIP can be used as a drop-in replacement for VQ in any rounding framework. In the following sections, we first describe the "bitshift" trellis (Section 3.1). Then, we describe a series of fast compute-based codes for i.i.d Gaussian sources, aligning with different types of hardware (Sections 3.1.1 and 3.1.2). Finally, we give an approximation for the tail-biting trellis problem, which lets us more efficiently load weights in hardware (Section 3.2).

### 3.1 "Bitshift" Trellis and Codebook Design

The bitshift trellis was introduced by Mao and Gray [23] as part of the "random permutation trellis coder" (RPTC). In the bitshift trellis, node $i$ has an edge to node $j$ if $\exists c \in \mathbb{Z}, 0 \leq c < 2^{kV}$, s.t. $j = (i2^{kV} \mod 2^L) + c$. Essentially, the top $L - kV$ bits of $j$ equal the bottom $L - kV$ bits of $i$. This means that the first group of $V$ weights depends only on the bits at positions $\{1, 2, \dots, L\}$, the second only on bit positions $\{kV + 1, kV + 2, \dots, kV + L\}$, and in general the $t$th on bit positions $\{(t-1)kV + 1, \dots, (t-1)kV + L\}$. During inference, obtaining the next compressed group of $V$ weights in a sequence only requires bitshifting by $kV$ bits, which is supported on virtually all hardware. Furthermore, since each group of $V$ weights only depends on a contiguous window of $L$ bits in $\hat{S}$, decoding can be parallelized. Figure 2 shows a simple ($L = 2, k = 1, V = 1$) bitshift trellis. Note that edges only exist between nodes that overlap by 1 bit, and storing the quantized length 6 $\hat{S}$ indeed only requires 6 bits (plus the initial state).

Quantizing an i.i.d. source with the bitshift trellis is nontrivial because neighboring groups of weights sharing many bits can potentially lead to strong correlations (Figure 3 LL). The RPTC permutes

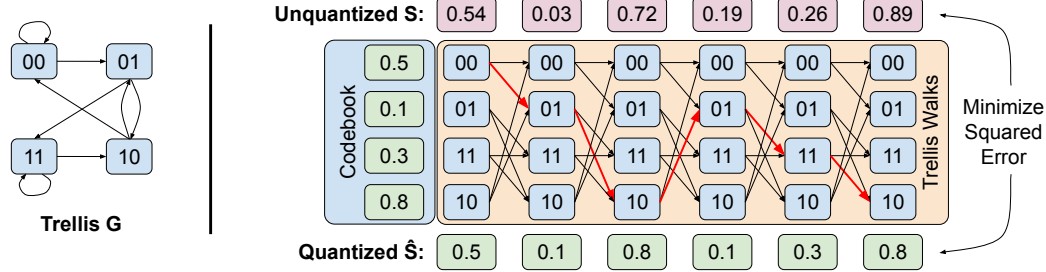

Figure 2: A bitshift trellis code with $L = 2, k = 1, V = 1$. Nodes 0, 1, 2, and 3 have code values 0.5, 0.1, 0.8, and 0.3, respectively. Each node can only transition to the $2^{kV} = 2$ nodes that share their top $L - kV = 1$ bit with its bottom $L - kV = 1$ bit. In this example, $\hat{S}$ can be stored as `0010110`. $\hat{S}$ is also *tail-biting*, so the last $L - kV = 1$ bits can be dropped to give $\hat{S} = $ `001011`.

Table 1: QTIP's compute-based codes (1MAD, 3INST, HYB) achieve similar distortion rates as a pure-lookup random Gaussian trellis code (RPTC) when quantizing an i.i.d Gaussian source to 2 bits. All TCQ methods ($L = 16$) outperform SQ and VQ and are significantly closer to the infinite-length distortion rate $D_R$, which lower bounds the distortion a $k$-bit quantizer can attain.

| | SQ | VQ | 1D TCQ | | | 2D TCQ | | |
|---|---|---|---|---|---|---|---|---|
| QUANT. | LLOYD-MAX | QUIP# E8P | 1MAD | 3INST | RPTC | HYB | RPTC | $D_R$ |
| DIM. | 1 | 8 | 256 | 256 | 256 | 256 | 256 | $\infty$ |
| MSE. | 0.118 | 0.089 | 0.069 | 0.069 | 0.068 | 0.071 | 0.069 | 0.063 |

the codebook to decorrelate neighboring weight groups (Figure 3 RR) [23]. However, this requires storing the codebook or storing and applying the permutation, both of which are prohibitively expensive during decoding. Instead, QTIP introduces a series of compute-based codes to produce a psuedorandom code, which has the same decorrelating effect and admits fast inference. To match approximately i.i.d. Gaussian RHT-transformed matrices, these codes produce psuedorandom approximate Gaussians in as few as 2 instructions per weight (see Table 1 and Figure 3). To the best of our knowledge, these code constructions alone are novel and we are the first to propose a lookup-free Gaussian trellis code.

### 3.1.1 Lookup-Free Computed Codes

Here, we present two pure-computed lookup-free codes that produce a pseudorandom approximately Gaussian number from a $L$ bit word, enabling fast decoding on cache-limited hardware. These codes avoid strong correlations and can be implemented in $\leq 4$ hardware instructions per weight on NVIDIA GPUs. We present two codes here to illustrate that multiple such codes are possible: in practice a lookup-free code can be designed to use the instructions available on whatever hardware we want to run on.

Algorithm 1 (1MAD) first runs a linear congruential generator (LCG) to produce a pseudorandom 32-bit word [32]. This requires 2 instructions (`MAD` and `&`). It then sums the 32-bit word as four 8-bit unsigned integers; this sum is approximately Gaussian distributed. This requires 1 instruction (`vabsdiff4`). Finally, this sum must be scaled and shifted (another `MAD`). Although there are only $2^{10}$ representable values even when $L > 10$, this does not empirically affect quantization quality. 1MAD requires choosing $a$ and $b$ to avoid strong correlations; we set $a = 34038481$ and $b = 76625530$ (Figure 3 LC).

Algorithm 2 (3INST) also first runs an LCG to produce a random 32-bit word $X$. Then, it XORs the bottom 16 bits of $X$ with the mantissa bits, bottom two exponent bits, and sign bit of a magic FP16 number $m$ to produce an FP16 number $m_1$. It then repeats this with the top 16 bits of $X$ to produce $m_2$ and returns $m_1 + m_2$. This entire process can be implemented in 3 ALU instructions[1] with a `MAD` for the LCG, a `lop3` to mask and XOR with a packed duplicated $m$, and then summing $m_1$ and $m_2$.

---

[1] As there is currently no instruction on NVIDIA GPUs that sums the top and bottom half of a 32-bit word as two FP16s, this requires an extra data movement instruction to "split" the 32-bit word into two 16-bit registers.

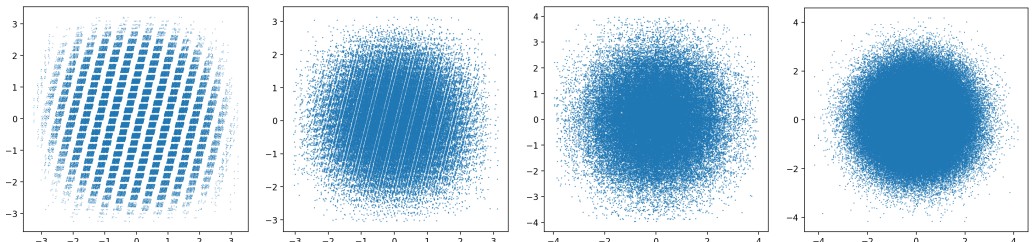

Figure 3: Set of representable neighboring values in a bitshift trellis with $L = 16, k = 2, V = 1$ for (far left) a code with strong correlations, (left center) algorithm 1 ("1MAD"), (right center) algorithm 2 ("3INST"), and (far right) a random Gaussian code. Note that while 1MAD has minor correlations, both 1MAD and 3INST are close to a random Gaussian, resulting in good quantization quality.

---

**Algorithm 1** Computed Gaussian Code "1MAD"

---

**input** $L$-bit 0 left-padded integer $x$, uint32 $a, b$.
  $x \leftarrow (ax + b) \bmod 2^{32}$ {run LCG to get uniform random $x$}
  {sum $x$ as four 8-bit unsigned integers, this is approximately Gaussian}
  $x \leftarrow (x \text{ \& } 255) + ((x \text{ >> } 8) \text{ \& } 255) + ((x \text{ >> } 16) \text{ \& } 255) + ((x \text{ >> } 24) \text{ \& } 255)$
  $x \leftarrow (x - 510)/147.8$
**output** Pseudorandom approximate Gaussian $x$.

---

$m_1 + m_2$ is approximately distributed by the sum of two mirrored exponential distributions, which is close to Gaussian. Like with Algorithm 1, $a, b$, and $m$ must be chosen to avoid correlations; we used $a = 89226354, b = 64248484, m = 0.922$ (Figure 3 right).

### 3.1.2 Hybrid Lookup-Computed Codes

Here, we describe a hybrid computed-lookup code that computes a pseudorandom (or hashed) index into a 2D vector codebook ($V = 2$). This code is tailored for modern GPUs, which have enough cache for a small in-memory LUT—one benefit of using such a LUT over a purely computed codebook is that a LUT can be fine-tuned after quantization. Algorithm 3 first performs the hash $X \leftarrow X^2 + X$ to mix the lower order and upper order bits of $X$ [18]. Then, it takes bits $(14 - Q + 1) - 14$ (0 indexed) as an index into a $2^Q \times 2$ LUT to get two 16-bit floats. (The reason why we chose a 2D codebook here is that shared memory on NVIDIA GPUs is accessed in 32-bit-word elements, and each such word can contain two 16-bit floats.) Finally, it XORs bit 15 of $X$ to flip the sign of the second entry of the codebook vector. Algorithm 3 can be implemented with MAD, bitshift, mask, and lop3, giving an amortized 2 instructions per weight. This effectively assigns a $L$ bit word to one of $2^{Q+1}$ 2D vectors, each of which can be fine-tuned to improve quality. Algorithm 3 can also be implemented to XOR bit 31 alongside bit 15 (this is free in the lop3) to give an effectively $2^{Q+2}$-sized codebook, which can improve quantization quality. We only realized this after running all the experiments, so the numbers in this paper use the "one sign flip" version of Algorithm 3. In QTIP, we initialize the LUT using K-means on an empirical 2D i.i.d. Gaussian distribution.

### 3.2 Tail-Biting Trellises

Directly quantizing a length-$T$ sequence to a $(L, k, V)$ trellis results in a total of $kT + L - kV$ bits since the starting state takes an additional $L - kV$ bits to store. If we run inference on a machine with $w$-bit words where $w|kT$, we must read an extra $\lceil \frac{L-kV}{w} \rceil w - (L - kV)$ wasted bits per sequence. For common $w$ (e.g. 32), setting $L = kV + w$ makes the Viterbi algorithm intractable. One way to solve this is by enforcing that the start and end state share $L - kV$ bits, i.e. the trellis is *tail-biting* [4]. Exactly solving the tail-biting trellis problem via dynamic programming takes time quadratic in the state space ($2^L$), making this problem intractable for reasonable $L \geq 12$ [30]. However, since RHT-processed weights are approximately i.i.d., simple algorithms can be effective for approximately solving the tail-biting problem. We propose Algorithm 4, which first rotates the sequence by $T/2$, quantizes it, and then extracts the overlap between the rotated start and end states. It then requantizes the original sequence with this overlap as the tail-biting overlap. This only requires two Viterbi calls

---

**Algorithm 2** Computed Gaussian Code "3INST"

---

**input** $L$-bit 0 left-padded integer $x$, `uint32` $a, b$, `float16` $m$.
   $x \leftarrow (ax + b)$ mod $2^{32}$ {run LCG to get uniform random $x$}
   {modify sign, mantissa, and bottom 2 exponent bits of $m$ and sum, this is approximately Gaussian}
   $m \leftarrow$ `reinterpret`$(m,$`uint32`$)$ `<< 16` $+$ `reinterpret(m, uint32)`
   $x \leftarrow (x$ `&` `b10001111111111111000111111111111`$)$ `XOR` $m$
   $x \leftarrow$ `reinterpret`$(x$ `&` $2^{16} - 1,$ `float16`$) +$ `reinterpret`$((x$ `>> 16`$)$ `&` $2^{16} - 1,$ `float16`$)$
**output** Pseudorandom approximate Gaussian $x$.

---

---

**Algorithm 3** Hybrid Computed-Lookup 2D Gaussian Code "HYB"

---

**input** $L$-bit 0 left-padded integer $x$, codebook $C \in \mathbb{R}^{2^Q \times (V=2)}$.
   $x \leftarrow x \cdot x + x$ mod $2^{32}$ {calculate hash}
   $v \in \mathbb{R}^2 \leftarrow C[(x$ `>>` $(15 - Q))$ `&` $2^Q - 1]$ {lookup from symmetric codebook}
   $v \leftarrow v$ `XOR` $(x$ `&` $(1$ `<< 15`$))$ {apply sign flip}
**output** Pseudorandom approximate Gaussian vector $v$.

---

in total. Table 2 shows that in practice, Algorithm 4 can find close-to-optimal tail-biting sequences while being significantly cheaper to run than other tail-biting approximation algorithms [30].

## 4 Experiments

Here, we present experiments quantizing the Llama family of models with QTIP [33, 34, 26]. These models offer strong performance across a wide range of sizes, allowing us to compare how different quantization methods perform and scale. We primarily compare QTIP against QuIP# and AQLM. For Llama 1, we include GPTVQ-2D instead of AQLM since AQLM does not publish Llama 1 numbers [36]. GPTVQ-2D performs 2D VQ inside GPTQ and offers strong performance. These methods outperform scalar quantization methods including GPTQ, AWQ, and OmniQuant; comparisons to those methods can be found in QuIP# and AQLM [20, 14, 29, 35, 11]. We mainly focus on the hybrid code (Section 4.2) since it is tailored for modern GPUs, and present a full suite of results for it. For the computed codes (Section 4.1), we present results for Llama 2.

Since the proxy error is not an additive distortion metric, we cannot minimize it by quantizing $W$ as one sequence. Instead, for all experiments, we use QTIP as a quantizer in QuIP#'s BlockLDLQ, which allows us to simultaneously achieve high dimensionality and low proxy error [35]. Specifically, we quantize a block of $T_x \times T_y$ weights as a sequence, where $T_x$ and $T_y$ span the output and input dimensions of $W$, respectively. Since BlockLDLQ only specifies feedback along the input dimension, this is equivalent to BlockLDLQ with $g = T_y$ but a vector dimension of $T_x T_y \gg T_y$. This has the benefit of limiting the effect of $g$ in BlockLDLQ's error bound $gm\mu^2\sigma^2\text{tr}(H^{1/2})^2/n$ while achieving a high dimension for TCQ. Algorithm 5 in the Appendix describes this in more detail.

### 4.1 Lookup-Free Computed Codes

Here, we use 1MAD and 3INST with $L = 16, V = 1, T_x = T_y = 16$. Setting $T_x = T_y = 16$ enables using a $16 \times 16$ MMA tile per trellis sequence to perform matrix multiplication during inference. $16 \times 16$ MMA tiles form the basis of many types of "AI hardware," making fast decoding relatively simple [6]. We do not perform fine-tuning since the codes themselves are not tunable, but these codes are fully compatible with QuIP#-style fine-tuning (recall that QuIP#'s codebook is also not tunable). Table 3 shows that both 1MAD and 3INST significantly outperform QuIP# without fine-tuning (AQLM does not have numbers without fine-tuning). Even at 4 bits, where all methods are close to lossless, QTIP results in significant improvements. Notably, the computed-code QTIP variants *without* fine-tuning outperforms both QuIP# and AQLM *with* fine-tuning on almost all models and sizes, showing that fine-tuning is not a silver bullet.

### 4.2 Hybrid Lookup-Computed Codes

**Algorithm 4** Tail-biting Trellis Approx.

**input** Sequence $S \in \mathbb{R}^T$, $(L, k, V)$ Trellis $G$.
  $S' \leftarrow$ Rotate $S$ to the right by $\lfloor T/2 \rfloor$
  $\hat{S}' \leftarrow$ Viterbi$(S', G)$
  $O \leftarrow L - kV$ bit overlap of $\hat{S}'_{\lfloor T/2 \rfloor} \hat{S}'_{\lfloor T/2 \rfloor + 1}$
  $\hat{S} \leftarrow$ Viterbi$(S, G)$ with start/end overlap $= O$
**output** Tail biting $\hat{S}$

Table 2: Quantizing 4K $T = 256$ i.i.d Gaussian seqs. with a tail-biting $(12, k, 1)$ trellis.

| $k$ | Alg. 4 MSE | Optimal MSE |
|---|---|---|
| 1 | 0.2803 | 0.2798 |
| 2 | 0.0733 | 0.0733 |
| 3 | 0.0198 | 0.0198 |
| 4 | 0.0055 | 0.0055 |

Table 3: Wikitext2 and C4 perplexity (↓), ctx. 4096, QTIP with pure-computed codes. Even *without fine-tuning*, pure-computed QTIP outperforms QuIP# and AQLM, both of which use fine-tuning, at almost all models sizes.

| | | 4 BIT NO FT | | | ≈4 BIT | | 3 BIT NO FT | | | ≈3 BIT | | 2 BIT NO FT | | | ≈2 BIT | |
|---|---|---|---|---|---|---|---|---|---|---|---|---|---|---|---|---|---|
| | | FP16 | 1MAD | 3INST | QuIP# | QuIP# | AQLM | 1MAD | 3INST | QuIP# | QuIP# | AQLM | 1MAD | 3INST | QuIP# | QuIP# | AQLM |
| 2-7 | W2 | 5.12 | **5.17** | **5.17** | 5.22 | 5.19 | 5.21 | **5.38** | 5.40 | 5.60 | 5.41 | 5.38 | 7.05 | **6.82** | 8.22 | 6.19 | 6.14 |
| | C4 | 6.63 | **6.71** | **6.71** | 6.79 | 6.75 | 6.75 | **6.99** | 7.01 | 7.34 | 7.04 | 7.01 | 9.14 | **8.96** | 11.0 | 8.16 | 8.09 |
| 2-13 | W2 | 4.57 | **4.62** | **4.62** | 4.65 | 4.63 | 4.64 | **4.74** | **4.74** | 4.90 | 4.78 | 4.78 | 5.59 | **5.52** | 6.06 | 5.35 | 5.33 |
| | C4 | 6.05 | **6.10** | **6.10** | 6.15 | 6.13 | 6.14 | **6.28** | **6.28** | 6.50 | 6.35 | 6.33 | 7.46 | **7.39** | 8.07 | 7.20 | 7.19 |
| 2-70 | W2 | 3.12 | **3.16** | **3.16** | 3.18 | 3.18 | 3.19 | **3.27** | **3.27** | 3.41 | 3.35 | 3.36 | **3.87** | 3.90 | 4.16 | 3.91 | 3.83 |
| | C4 | 4.97 | **5.00** | **5.00** | 5.02 | 5.02 | 5.03 | **5.09** | **5.09** | 5.20 | 5.15 | 5.17 | 5.70 | **5.69** | 6.01 | 5.71 | 5.62 |

Here, we use the hybrid lookup-computed code with $L = 16, V = 2, T_x = T_y = 16, Q = 9$. Setting $Q = 9$ gives a 2KiB codebook, which fits in L1 cache *even after* duplication for bank conflicts ($32\times$) on modern GPUs. This codebook is differentiable, so we can fine-tune it: to evaluate this, we fine-tune using QuIP#'s methodology, tuning both the codebook entries and the as-yet-unquantized weights in a blockwise fashion. Table 5 shows the perplexity of quantized Llama 1 and 2 models. In all cases, QTIP outperforms the other vector quantization-based methods. Even at 3 and 4 bits, where QuIP# and AQLM are close to lossless,

Table 4: Batch size 1 decoding throughput on a RTX6000 Ada (960GB/s mem. BW).

| METHOD | BITS | 2-7B TOK/S | 2-70B TOK/S |
|---|---|---|---|
| FP16 | 16 | 55.9 | OOM |
| AQLM | 2 | 81.5 | 8.78 |
| QuIP# | 2 | 186 | 22.2 |
| QTIP | 2 | 188 | 23.5 |
| QTIP | 3 | 161 | 19.1 |
| QTIP | 4 | 140 | 16.3 |

QTIP roughly *halves* the perplexity gap. These results also show the importance of dimensionality. Note that the 3- and 4-bit Llama 2 70B numbers here match those in 3. Since Table 3 uses a pure-computed code *without fine-tuning*, fine-tuning has no effect in these regimes and the improvement over QuIP# is purely from dimensionality.

Table 6 shows zeroshot results computed with LM Eval, which are slightly random; QTIP generally matches or exceeds QuIP# and AQLM on these tasks [15]. Table 7 contains results on Llama 3. Like other works, we have observed that Llama 3 (especially 70B base) is harder to quantize than Llama 2 [16]. Since the contribution and focus of this work is *what to round with* (TCQ) and not *how to round* (BlockLDLQ), we only compare against the proximal baseline QuIP#, which uses BlockLDLQ with VQ. QTIP significantly improves upon QuIP# at all model sizes and bitrates, once again showing the dimensionality advantage of TCQ over VQ. Table 8 shows results for Llama 3.1 instruct-tuned models, including Llama 3.1 405B. At all sizes, QTIP achieves strong results. Notably, QTIP is able to match or exceed PV-Tuning, a recent quantization method that focuses on better fine-tuning algorithms [22]. However, PV-Tuning is based off of AQLM and inherits its slow inference speed, making it significantly slower than QTIP. Finally, Table 9 shows results for quantizing Llama 3.2 instruct-tuned models to 4 bits. Since the embedding layers are very large relative to the decoder layers for small Llama 3 models ($\approx 500 - 750$MB), quantizing the decoder layers to fewer than 4 bits does not make a significant difference on the final model size. Here, QTIP is still able to achieve a meaningful end-to-end compression rate (2.5-3X) without degrading the final model.

Table 5: Wikitext2 and C4 perplexity (↓), QTIP with the hybrid-computed code. QTIP enables high-dimensional quantization and outperforms state-of-the-art vector quantization approaches.

| | | CTX. 2048, X = GPTVQ, Y = 0.13 | | | | | | | | CTX. 4096, X = AQLM, Y ≈ 0 | | | | | |
| | | WIKTEXT2 | | | | C4 | | | | WIKITEXT2 | | | C4 | | |
| METHOD | BITS | 1-7 | 1-13 | 1-30 | 1-65 | 1-7 | 1-13 | 1-30 | 1-65 | 2-7 | 2-13 | 2-70 | 2-7 | 2-13 | 2-70 |
|---|---|---|---|---|---|---|---|---|---|---|---|---|---|---|---|
| FP16 | 16.0 | 5.68 | 5.09 | 4.10 | 3.53 | 7.04 | 6.61 | 5.98 | 5.62 | 5.12 | 4.57 | 3.12 | 6.63 | 6.05 | 4.97 |
| X | 4+Y | 5.94 | 5.20 | 4.18 | 3.64 | – | – | – | – | 5.21 | 4.65 | 3.19 | 6.75 | 6.14 | 5.03 |
| QuIP# | 4.00 | 5.76 | 5.17 | 4.18 | 3.60 | 7.18 | 6.67 | 6.03 | 5.66 | 5.19 | 4.63 | 3.18 | 6.75 | 6.13 | 5.02 |
| QTIP | 4.00 | **5.72** | **5.15** | **4.15** | **3.58** | **7.13** | **6.65** | **6.01** | **5.64** | **5.17** | **4.61** | **3.16** | **6.69** | **6.09** | **5.00** |
| X | 3+Y | 6.32 | 5.31 | 4.38 | 3.79 | – | – | – | – | 5.38 | 4.78 | 3.36 | 7.01 | 6.33 | 5.17 |
| QuIP# | 3.00 | 5.98 | 5.31 | 4.36 | 3.70 | 7.39 | 6.83 | 6.17 | 5.77 | 5.41 | 4.78 | 3.35 | 7.04 | 6.35 | 5.15 |
| QTIP | 3.00 | **5.85** | **5.24** | **4.26** | **3.68** | **7.26** | **6.74** | **6.09** | **5.71** | **5.28** | **4.69** | **3.26** | **6.87** | **6.22** | **5.08** |
| X | 2+Y | 9.64 | 6.58 | 5.63 | 4.91 | – | – | – | – | 6.14 | 5.33 | 3.83 | 8.09 | 7.19 | 5.62 |
| QuIP# | 2.00 | 6.86 | 5.97 | 5.02 | 4.36 | 8.36 | 7.48 | 6.71 | 6.19 | 6.19 | 5.35 | 3.91 | 8.16 | 7.20 | 5.71 |
| QTIP | 2.00 | **6.52** | **5.80** | **4.83** | **4.21** | **7.99** | **7.31** | **6.56** | **6.08** | **5.86** | **5.11** | **3.70** | **7.73** | **6.85** | **5.48** |

Table 6: Zeroshot accuracy (↑), QTIP with the hybrid-computed code.

| | 2-70 | | | | | 2-13 | | | | | 2-7 | | | | |
| MTHD. | BITS | ARCC | ARCE | PIQA | WINO | BITS | ARCC | ARCE | PIQA | WINO | BITS | ARCC | ARCE | PIQA | WINO |
|---|---|---|---|---|---|---|---|---|---|---|---|---|---|---|---|
| FP16 | 16 | 51.1 | 77.7 | 81.1 | 77.0 | 16 | 45.6 | 73.3 | 73.5 | 69.6 | 16 | 40.0 | 69.3 | 78.5 | 67.3 |
| AQLM | 4.14 | 50.7 | 77.3 | 81.5 | 76.5 | 3.94 | **44.8** | 73.3 | 78.4 | **69.9** | 4.04 | 41.0 | 70.2 | 78.2 | 67.3 |
| QuIP# | 4 | 50.5 | 77.7 | 81.4 | **77.3** | 4 | 43.6 | 71.3 | 78.7 | 69.6 | 4 | 40.4 | 68.6 | **78.5** | **67.4** |
| QTIP | 4 | 50.0 | **77.8** | 81.3 | 76.9 | 4 | 44.8 | 73.6 | **78.9** | 69.9 | 4 | 40.0 | **68.9** | 78.4 | 67.1 |
| AQLM | 3.01 | 50.3 | 78.0 | 80.7 | 75.3 | 3.03 | 42.8 | 72.9 | 78.5 | 68.8 | 3.04 | 38.5 | 66.8 | 77.3 | 65.4 |
| QuIP# | 3 | **50.9** | 77.6 | **81.4** | 76.1 | 3 | **44.0** | 72.5 | 78.4 | 69.1 | 3 | **39.2** | **68.4** | 77.3 | 66.5 |
| QTIP | 3 | 50.3 | **78.2** | 80.6 | 77.0 | 3 | **44.0** | 72.8 | 78.0 | **69.5** | 3 | 38.9 | 68.1 | **78.1** | **66.9** |
| AQLM | 2.07 | 47.9 | **77.7** | 80.4 | 75.9 | 1.97 | 38.8 | 69.3 | 75.9 | 68.8 | 2.02 | 32.8 | 63.7 | 74.8 | 65.7 |
| QuIP# | 2 | 47.6 | 77.1 | 79.5 | 74.6 | 2 | 39.6 | 69.0 | **77.3** | 67.4 | 2 | 35.2 | 65.3 | 75.4 | **64.9** |
| QTIP | 2 | **48.0** | 76.3 | 80.2 | 75.1 | 2 | **41.4** | **70.8** | **77.3** | **67.6** | 2 | **35.7** | **65.6** | **75.9** | 64.7 |

## 4.3 Inference Speed

Table 4 shows the batch size 1 inference speed of QTIP, QuIP#, and AQLM on Llama 2 7B and 70B with matrix fusion. Here, the design choices of QTIP and QuIP# become apparent. Whereas AQLM uses a codebook that is too large to fit in cache and thus prevents fast inference, both QTIP and QuIP# achieve significant speedups over FP16. Furthermore, while it is impressive that both QuIP# and QTIP are > 2× faster than AQLM, it is even more impressive that QTIP is able to match QuIP#'s throughput with an effective dimension size of 256, or 32× larger than QuIP#'s. This means that the improved quantization quality of QTIP comes with *no additional inference-time cost*. Although our empirical throughput numbers were timed on NVIDIA GPUs, QTIP can be fast on a broad class of accelerators due to its flexibility. QTIP only requires generating a pseudorandom Gaussian efficiently, and can work on devices with no cache as well as devices with lookup hardware. For example, if we were using a ARMv8 CPU, we could use the `vqtbl4q_u8` NEON intrinsic to look up 16 indices in a 64-entry codebook. This would let us use a 6 bit 1D codebook with the HYB code (Q=6, V=1). Quantizing Llama 2 7B to 2 bits with this setup and w/out fine-tuning gives 6.89 Wikitext2 perplexity – essentially the same state-of-the-art quality as 3INST.

## 5 Conclusion

We present QTIP, a weight-only post-training quantization algorithm that achieves state-of-the-art results through the use of trellis-coded quantization (TCQ). TCQ enables tractable ultra-high dimensional quantization, significantly reducing quantization distortion over vector quantization (VQ). However, naive TCQ does not admit fast inference due to sequential bottlenecks during decoding and needing to store a large codebook. QTIP solves this problem through a novel combination of incoherence processing, the hardware-efficient bitshift trellis, and fast computed codes. Specifically, QTIP introduces a series of compute-based pseudorandom Gaussian codes that, when used in

Table 7: QTIP vs. QuIP#, Llama 3 (ctx. 8192 for perplexity). Although Hessian-based rounding generally underperforms on Llama 3, the focus of this work is on *what to quantize with* (TCQ vs. VQ). Here, the high-dimensionality of TCQ in QTIP improves over the low-dimensional VQ in QuIP#.

| | | 3-70 PPL (↓) | | 3-70 ZEROSHOT ACC (↑) | | | | | 3-8 PPL (↓) | | 3-8 ZEROSHOT ACC (↑) | | | | |
|---|---|---|---|---|---|---|---|---|---|---|---|---|---|---|---|
| Mthd. | Bits | W2 | C4 | ArcC | ArcE | BoolQ | PiQA | Wino | W2 | C4 | ArcC | ArcE | BoolQ | PiQA | Wino |
| BF16 | 16.0 | 2.59 | 5.78 | 60.5 | 86.9 | 85.3 | 82.4 | 80.3 | 5.54 | 7.10 | 50.2 | 80.1 | 81.0 | 79.7 | 72.9 |
| QuIP# | 4.00 | 2.99 | 5.96 | 35.0 | 67.3 | 84.7 | 71.9 | 76.7 | 5.81 | 7.32 | 50.2 | **79.7** | **81.3** | 79.7 | 73.1 |
| QTIP | 4.00 | **2.75** | 5.83 | **56.1** | **83.9** | **85.8** | **81.3** | **80.6** | **5.67** | **7.20** | 50.2 | 79.6 | 79.5 | 79.4 | **73.4** |
| QuIP# | 3.00 | 3.59 | 6.18 | 31.1 | 36.6 | **85.7** | 58.8 | 76.4 | 6.27 | 7.71 | 46.4 | 77.4 | 79.9 | 77.9 | 72.9 |
| QTIP | 3.00 | **3.18** | 5.98 | **48.6** | **77.8** | 85.0 | **77.8** | **79.7** | **6.01** | **7.48** | **49.2** | **79.3** | **80.0** | **79.2** | **74.5** |
| QuIP# | 2.00 | 5.77 | 7.46 | 18.3 | 32.2 | 82.1 | 54.7 | 68.9 | 7.84 | 9.06 | 39.2 | 72.9 | 76.6 | 75.6 | 68.2 |
| QTIP | 2.00 | **4.97** | **6.80** | **28.0** | **35.2** | **83.6** | **57.1** | **72.6** | **7.33** | **8.62** | **44.2** | **75.2** | **76.7** | **77.6** | **70.7** |

Table 8: Llama 3.1 instruct-tuned model results (ctx. 8192 for perplexity). QTIP performs well at all model sizes and generally outperforms PV-Tuning, a recent quantization method that focuses on fine-tuning algorithms. The zeroshot results in this table use LM Eval 0.4.4 and the "standard" versions of each task instead of the Meta versions in [26].

| | | | Ppl. (↓) | Zeroshot (↑) | | | |
|---|---|---|---|---|---|---|---|
| | | Bits | W2 | ArcC | ArcE | Hswag | PiQA |
| 3.1 405B Inst. | Meta "FP8" | 16 Attn. / 8 MLP | 1.70 | 61.6 | 81.4 | 67.1 | 83.8 |
| | QTIP | 4 | 1.79 | 61.3 | 80.9 | 66.7 | 84.2 |
| | QTIP | 3 | 2.05 | 61.5 | 81.4 | 66.8 | 83.5 |
| | QTIP | 2 | 3.29 | 60.7 | 81.1 | 65.4 | 82.2 |
| 3.1 70B Inst. | BF16 | 16 | 3.52 | 56.7 | 75.6 | 61.5 | 82.8 |
| | QTIP | 4 | 3.73 | 56.3 | 75.8 | 61.4 | 83.0 |
| | QTIP | 3 | 4.12 | 55.1 | 75.1 | 60.8 | 82.6 |
| | QTIP | 2 | 5.08 | 54.4 | 72.6 | 59.4 | 82.5 |
| | PV-Tuning | 2.01 | 5.70 | 52.7 | 72.2 | 60.2 | 82.6 |
| 3.1 8B Inst. | BF16 | 16 | 6.50 | 51.6 | 77.8 | 57.7 | 80.0 |
| | QTIP | 4 | 6.61 | 50.7 | 78.0 | 57.5 | 80.1 |
| | QTIP | 3 | 6.80 | 50.4 | 77.7 | 56.9 | 79.3 |
| | QTIP | 2 | 7.82 | 45.1 | 75.6 | 54.5 | 79.0 |
| | PV-Tuning | 2.07 | 8.45 | 46.2 | 75.4 | 54.4 | 78.7 |

Table 9: Llama 3.2 instruct-tuned results when quantizing to 4 bits (ctx. 8192 for perplexity). Even on extremely small models, QTIP is still able to achieve meaningful compression without sacrificing quality. This table uses the same LM Eval setup as Table 8.

| | | Ppl (↓) | | Zeroshot (↑) | | | |
|---|---|---|---|---|---|---|---|
| | | Size (GB) | W2 | ArcC | ArcE | Hswag | PiQA |
| 3B | BF16 | 6 | 9.58 | 43.3 | 74.3 | 52.2 | 75.7 |
| | QTIP | 2.1 | 9.77 | 43.5 | 74.3 | 51.9 | 75.1 |
| 1B | BF16 | 2.4 | 11.57 | 36.0 | 68.5 | 45.2 | 74.2 |
| | QTIP | 0.97 | 11.93 | 34.8 | 68.4 | 44.5 | 73.3 |

conjunction with the bitshift trellis and incoherence processing, simultaneously achieves state-of-the-art PTQ quality and fast inference. QTIP improves quantization quality at all tested bitrates over the latest VQ-based PTQ methods, QuIP# and AQLM, further pushing the boundary of LLM PTQ. QTIP's codes use as few as 2 instructions per weight during decoding, enabling matrix-vector multiplication to run at over 80% of peak memory bandwidth on modern GPUs. Altogether, our results indicate that high dimensional quantization is necessary for high-quality compression, and QTIP is the first LLM PTQ method to scale to ultra-high dimensions while supporting fast inference.

## Acknowledgements

C.D. was supported by NSF-2046760 CAREER. We thank Together AI for compute resources.

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

# A   Appendix

## A.1   Additional Results

### A.1.1   Ablations on Trellis Size

Table 10 shows an ablation on $L$ for quantizing Llama 2 7B with $K = 2, V = 1$, the bitshift trellis, a pure-lookup codebook, and no fine-tuning. $L = 8$ is the largest $L$ achievable if we had to store the trellis and codebook in the same amount of cache as the HYB code (2KiB). $L = 10$ is the largest $L$ achievable if we only had to store the codebook. As expected, increasing $L$ improves quality. Table 10 also shows very little difference between an equal-sized LUT codebook and QTIP's codes, meaning that QTIP isn't sacrificing quality for speed. However, an equal-sized LUT would need $> 10\times$ more cache than the latest GPUs have, making the bitshift trellis and compute-based codes necessary to achieve both quality and speed. Table 11 shows an ablation on $V$ with $L = 12$ and 16, $K = 2$, and the same settings as Table 10. Increasing $V$ generally decreases quality, but this can be recovered with a larger $L$. It is hard to measure $V$'s impact on decoding speed since this is highly implementation and hardware dependent, so $V$ is more of a user-chosen hyperparameter.

Table 10: Ablation on $L$ when quantizing Llama 2 7B to 2 bits ($K = 2$ and $V = 1$).

| L | Trellis Size | CB size | total size | W2 | C4 |
|---|---|---|---|---|---|
| QuIP# | - | 8Kb | 8Kb | 8.22 | 11.0 |
| 8 | 8.19 Kb | 4.10 Kb | 12.29 Kb | 7.83 | 10.3 |
| 10 | 40.96 Kb | 16.38 Kb | 57.34 Kb | 7.49 | 9.67 |
| 12 | 196.61 Kb | 65.54 Kb | 262.14 Kb | 6.97 | 9.21 |
| 16 | 4.19 Mb | 1.05 Mb | 5.24 Mb | 6.83 | 8.92 |
| 16 | Bitshift | 3INST | 0Kb | 6.82 | 8.96 |

Table 11: Ablation on $V$ when quantizing Llama 2 7B to 2 bits ($K = 2$).

| Codebook | L | V | W2 | C4 |
|---|---|---|---|---|
| LUT | 12 | 1 | 6.97 | 9.21 |
| LUT | 12 | 2 | 7.09 | 9.24 |
| LUT | 12 | 4 | 7.55 | 9.88 |
| LUT | 16 | 1 | 6.83 | 8.92 |
| LUT | 16 | 2 | 6.79 | 8.97 |
| QTIP HYB (no FT) | 16 | 2 | 6.83 | 8.97 |
| LUT | 16 | 4 | 6.92 | 9.07 |

### A.1.2   Zeroshot Results

Table 12: Zeroshot results for the 1MAD code.

| | Bits | ArcC (acc) | ArcE (acc) | BoolQ (acc) | PiQA (acc) | Wino (acc) |
|---|---|---|---|---|---|---|
| 2-7 | 16 | 39.9 | 69.3 | 71.1 | 78.4 | 67.2 |
| 2-7 | 4 | 39.0 | 69.4 | 72.0 | 78.4 | 67.9 |
| 2-7 | 3 | 38.8 | 68.0 | 68.2 | 77.6 | 68.4 |
| 2-7 | 2 | 32.1 | 63.5 | 66.3 | 73.3 | 62.7 |
| 2-13 | 16 | 45.6 | 73.3 | 69.1 | 78.7 | 69.7 |
| 2-13 | 4 | 45.6 | 72.9 | 68.1 | 78.7 | 70.3 |
| 2-13 | 3 | 42.2 | 71.0 | 69.9 | 78.6 | 69.8 |
| 2-13 | 2 | 38.5 | 71.5 | 71.4 | 75.9 | 68.9 |
| 2-70 | 16 | 51.2 | 77.7 | 76.7 | 81.1 | 76.9 |
| 2-70 | 4 | 51.1 | 77.8 | 75.2 | 81.5 | 77.0 |
| 2-70 | 3 | 50.8 | 77.8 | 77.9 | 80.7 | 76.3 |
| 2-70 | 2 | 49.3 | 77.7 | 83.3 | 80.4 | 75.7 |

Table 13: Zeroshot results for the 3INST code.

| | Bits | ArcC (acc) | ArcE (acc) | BoolQ (acc) | PiQA (acc) | Wino (acc) |
|---|---|---|---|---|---|---|
| 2-7 | 16 | 39.9 | 69.3 | 71.1 | 78.4 | 67.2 |
| 2-7 | 4 | 40.2 | 68.5 | 70.3 | 78.0 | 67.7 |
| 2-7 | 3 | 40.2 | 68.6 | 73.0 | 77.5 | 65.4 |
| 2-7 | 2 | 32.9 | 61.9 | 65.5 | 74.5 | 65.0 |
| 2-13 | 16 | 45.6 | 73.3 | 69.1 | 78.7 | 69.7 |
| 2-13 | 4 | 45.4 | 72.7 | 67.9 | 78.5 | 69.9 |
| 2-13 | 3 | 44.5 | 72.6 | 70.1 | 78.5 | 69.4 |
| 2-13 | 2 | 38.7 | 68.2 | 63.6 | 75.6 | 68.7 |
| 2-70 | 16 | 51.2 | 77.7 | 76.7 | 81.1 | 76.9 |
| 2-70 | 4 | 50.3 | 77.9 | 77.3 | 80.7 | 76.5 |
| 2-70 | 3 | 50.9 | 78.3 | 78.8 | 81.1 | 77.5 |
| 2-70 | 2 | 48.0 | 76.5 | 76.7 | 80.1 | 77.6 |

Table 14: Llama 1 Zeroshot results for the Hybrid code

| | Bits | ArcC (acc) | ArcE (acc) | BoolQ (acc) | PiQA (acc) | Wino (acc) |
|---|---|---|---|---|---|---|
| 1-7 | 16 | 38.2 | 67.4 | 73.1 | 78.4 | 67.0 |
| 1-7 | 4 | 38.8 | 67.1 | 74.2 | 78.3 | 67.1 |
| 1-7 | 3 | 37.0 | 65.7 | 74.1 | 77.7 | 67.3 |
| 1-7 | 2 | 35.3 | 64.9 | 72.9 | 76.1 | 65.4 |
| 1-13 | 16 | 43.9 | 74.6 | 68.5 | 78.8 | 70.1 |
| 1-13 | 4 | 43.4 | 73.7 | 68.2 | 79.1 | 70.1 |
| 1-13 | 3 | 42.2 | 74.2 | 68.0 | 78.7 | 70.5 |
| 1-13 | 2 | 39.7 | 72.1 | 66.6 | 77.6 | 68.9 |
| 1-30 | 16 | 46.7 | 75.4 | 68.4 | 81.0 | 72.6 |
| 1-30 | 4 | 46.7 | 75.4 | 69.9 | 81.0 | 73.3 |
| 1-30 | 3 | 47.8 | 75.0 | 70.0 | 80.4 | 73.6 |
| 1-30 | 2 | 44.0 | 72.7 | 72.8 | 78.7 | 71.7 |
| 1-65 | 16 | 47.0 | 75.3 | 82.3 | 81.5 | 77.2 |
| 1-65 | 4 | 46.8 | 74.5 | 82.8 | 81.4 | 76.6 |
| 1-65 | 3 | 46.8 | 75.3 | 83.0 | 81.3 | 75.9 |
| 1-65 | 2 | 44.4 | 74.2 | 83.1 | 80.4 | 75.7 |

### A.1.3 Lookup-Only Codes

Table 15: Wikitext2 and C4 perplexity ($\downarrow$), ctx. 4096, QTIP with a size $2^{14}$ LUT codebook. This codebook is too large (32KB) for current GPU L1 caches, but could fit on near-future hardware.

| | | | ~4 Bit | | | ~3 Bit | | | ~2 Bit | | |
|---|---|---|---|---|---|---|---|---|---|---|---|
| | | FP16 | QTIP | QuIP# | AQLM | QTIP | QuIP# | AQLM | QTIP | QuIP# | AQLM |
| 2-7 | W2 | 5.12 | **5.16** | 5.19 | 5.21 | **5.30** | 5.41 | 5.46 | **5.89** | 6.19 | 6.64 |
| | C4 | 6.63 | **6.68** | 6.75 | 6.75 | **6.86** | 7.04 | 7.08 | **7.78** | 8.16 | 8.56 |
| 2-70 | W2 | 3.12 | **3.15** | 3.18 | 3.19 | **3.26** | 3.35 | 3.36 | **3.77** | 3.91 | 3.94 |
| | C4 | 4.97 | **4.99** | 5.02 | 5.03 | **5.07** | 5.15 | 5.17 | **5.55** | 5.71 | 5.72 |

Here, we use a pure-lookup code $\sim \mathcal{N}(0,1)$ with $L = 14, V = 1, T_x = 32, T_y = 8$, and QuIP#'s fine-tuning scheme. These parameters show what performance QTIP could achieve if we did not care about fast inference *today*. Specifically, a pure-lookup codebook is tunable, and setting $T_y = 8$ reduces the BlockLDLQ group size while maintaining high dimensionality (256). This codebook uses 32KB; this only fits in GPU L1 cache with bank conflicts. Setting $T_x = 32, T_y = 8$ corresponds to using a larger MMA tile size than current GPUs allow for. The largest tile size is usually 16 in the $T_x$ dimension, meaning that a $32 \times 8$ trellis needs two tiles. Thankfully, hardware required to serve

Table 16: Wikitext2 and C4 zeroshot accuracy (↑), QTIP with a size $2^{14}$ LUT codebook. This codebook is too large (32KB) for current GPU L1 caches, but could fit on near-future hardware.

|  | Bits | ArcC (acc) | ArcE (acc) | BoolQ (acc) | PiQA (acc) | Wino (acc) |
|---|---|---|---|---|---|---|
| 2-7 | 16 | 40.0 | 69.3 | 71.0 | 78.5 | 67.3 |
| 2-7 | 4 | 40.3 | 69.2 | 73.0 | 78.1 | 67.5 |
| 2-7 | 3 | 39.1 | 69.3 | 69.6 | 77.8 | 66.3 |
| 2-7 | 2 | 37.0 | 64.6 | 67.2 | 75.6 | 66.9 |
| 2-70 | 16 | 51.1 | 77.7 | 76.6 | 81.1 | 77.0 |
| 2-70 | 4 | 50.1 | 77.5 | 76.4 | 81.3 | 77.3 |
| 2-70 | 3 | 50.6 | 77.9 | 78.0 | 81.1 | 76.1 |
| 2-70 | 2 | 47.1 | 76.9 | 79.5 | 80.1 | 76.3 |

such a model quickly is likely only a few years away, as these parameters are only slightly outside of what today's hardware is capable of.

Table 15 shows that QTIP outperforms both QuIP# and AQLM at all compression ratios, with 3 bit QTIP achieving similar quality as 4 bit AQLM. While it is not fair to compare this QTIP setup with QuIP#, since QuIP# was designed for fast inference, we note that AQLM's VQ codebook uses $2^{16} \times 8 \times 2 = 1$ MiB. This is **32 times** larger than the QTIP codebook here, and would require 32 MiB of L1 cache to read from without bank conflicts. Not only is this orders of magnitude larger than current L1 caches (256KB on the H100), it is even larger than many **L2 caches!**

### A.1.4 Decoding Speed on Different GPUs

Table 17: Decoding speed on different Ampere and Lovelace GPUs.

| GPU Model | Model | 2-bit tok/s | 3-bit tok/s | 4-bit tok/s | FP16 tok/s |
|---|---|---|---|---|---|
| RTX 3090 | 2-7 | 127 | 119 | 109 | 52.5 |
| RTX 3090 | 2-70 | 15.3 | OOM | OOM | OOM |
| RTX A6000 Ampere | 2-7 | 116 | 106 | 95 | 43.5 |
| RTX A6000 Ampere | 2-70 | 15.0 | 13.1 | 11.7 | OOM |
| RTX 6000 Ada | 2-7 | 188 | 161 | 140 | 55.9 |
| RTX 6000 Ada | 2-70 | 23.5 | 19.1 | 16.3 | OOM |

### A.2 QTIP with BlockLDLQ

Here, we detail how we use TCQ within BlockLDLQ to produce our experimental setup. Essentially, QTIP is used as a high dimensional $T_x T_y$ quantizer within BlockLDLQ and is a drop-in replacement for vector quantization in BlockLDLQ. The regular blockLDLQ step $Q(W + (W - \hat{W})A)$ is exactly the same, and the only difference is in how $Q$ rounds. Instead of rounding each row of $x = W + (W - \hat{W})A$ independently, it groups $T_x$ rows into a block to round as $m/T_x$ high-dimensional sequences.

### A.3 Implementation Details

### A.3.1 Code

Our code is available at `https://github.com/Cornell-RelaxML/qtip`.

### A.3.2 Hessian Generation

Hessian matrices were generated with 6144 sequences of length 2048 for Llama 1, 6144 sequences of length 2048 for Llama 2, 4096 sequences of 8192 for Llama 3, and 4096 sequences of 8192 for Llama 3.1 except for 405B, which only used 2048 sequences due to time constraints. All sequences were sampled from the RedPajama dataset [7].

**Algorithm 5** QTIP with BlockLDLQ

**input** $W \in \mathbb{R}^{m \times n}, H \in \mathbb{R}^{n \times n}, T_x, T_y, L, k, V$, code $C$.

$\quad \hat{W} \leftarrow 0_{m,n}$
$\quad LDL^T \leftarrow T_y\text{-block LDL decomposition of } H$
$\quad A \leftarrow L - I$
$\quad \textbf{for } j \in \{n/T_y - 1, n/T_y - 2, ..., 0\} \textbf{ do}$
$\quad\quad x \leftarrow W_{:,jT_y:(j+1)T_y} + (W_{:,jT_y:} - \hat{W}_{:,jT_y:})A_{jT_y:,jT_y:(j+1)T_y}$
$\quad\quad x \leftarrow x.\text{reshape}(m/T_x, T_x T_y)$
$\quad\quad \hat{x} \leftarrow \text{Viterbi}(x, (L, k, V) \text{ bitshift trellis}, C) \text{ (row-wise)}$
$\quad\quad \hat{W}_{:,jT_y:(j+1)T_y} \leftarrow \hat{x}.\text{reshape}(m, T_y)$
$\quad \textbf{end for}$
**output** Quantized $\hat{W}$.

### A.3.3 Hadamard Matrices

We use Hadamard matrices from Neil Sloane's website [31].

### A.3.4 Perplexity

We use the OPTQ "Wikitext2" and "C4" test splits to calculate perplexity [14].

### A.3.5 Fine Tuning

We use the same fine-tuning hyperparemters as QuIP#, except that we do not perform matrix fusion [35]. This results in having to quantize 7 matrices (q, k, v, o, up, gate, down) instead of 4 (qkv, o, upgate, down). We quantize and fine tune in the following order: (v, q, k, o, up gate, down).

### A.4 Example Generations

The following are some sample generations from Llama 2 70B (not instruction tuned) quantized to 2 bits with QTIP, truncated to 256 tokens.

> **Prompt:** It is a truth universally acknowledged
>
> **QTIP:** It is a truth universally acknowledged that a single man in possession of a good fortune must be in want of a wife. However little known the feelings or views of such a man may be on his first entering a neighbourhood, this truth is so well fixed in the minds of the surrounding residents, that he is considered the rightful property of some one or other of their daughters. "My dear Mr. Bennet," said his lady to him one day, "have you heard that Netherfield Park is let at last?" "But it is," returned she; "for Mrs. Long has just been here, and she told me that Netherfield is taken by a young man of large fortune from the north of England; that he came down on Monday in a chaise and four to see the place, and was so much delighted with it that he agreed with Mr. Morris immediately; that he is to take possession before Michaelmas, and that he is now in town in search of a wife." "I see no occasion for that. You and the girls may go, or you may send them by themselves, which do you prefer? Only let me know what you are meaning to do, for I shall certainly not be at home. You

**Prompt:** Call me Ishamel

**QTIP:** Call me Ishmael. Some years ago—never mind how long precisely—having little or no money in my purse, and nothing particular to interest me on shore, I thought I would sail about a little and see the watery part of the world. It is a way I have of driving off the spleen, and regulating the circulation. Whenever I find myself growing grim about the mouth; whenever it is a damp, drizzly November in my soul; whenever I find myself involuntarily pausing before coffin warehouses, and bringing up the rear of every funeral I meet; and especially whenever my hypochondriacal palate is out of sorts, and I am desperately dull, and in the dumps, and have such an overpowering sense of fatality that I find myself involuntarily gravitating toward Cemeteries—and especially toward those open ones which are occasionally cheered by the gleam of a bonfire, or a cigar, or a pipe, or a lantern, or a candle, or a torch, or the moon, I then take to the ship. There is nothing surprising in this,

The following is a sample generation from Llama 3.1 405B Instruct quantized to 2 bits with QTIP.

**Prompt:** Give a proof for why the square root of 2 is irrational:

**QTIP:** The proof that the square root of 2 is irrational is as follows: Assume, for the sake of contradiction, that $\sqrt{2}$ is rational. This means it can be expressed as the ratio of two integers, $a$ and $b$, where $b$ is non-zero:

$$\sqrt{2} = a/b$$

We can further assume that $a$ and $b$ are not both even (i.e., they do not share a common factor of 2), because if they were, we could divide both by 2 and obtain a smaller representation of $\sqrt{2}$. Squaring both sides of the equation, we get:

$$2 = a^2/b^2$$

Multiplying both sides by $b^2$, we get:

$$2b^2 = a^2$$

This means that $a^2$ is even, because it is equal to 2 times an integer ($b^2$). Therefore, $a$ must also be even, because the square of an odd number is always odd. Since $a$ is even, we can write $a = 2c$, where $c$ is an integer. Substituting this into the equation above, we get:

$$2b^2 = (2c)^2$$

Expanding the right-hand side, we get:

$$2b^2 = 4c^2$$

Dividing both sides by 2, we get:

$$b^2 = 2c^2$$

This means that $b^2$ is even, and therefore $b$ is also even. However, this contradicts our assumption that $a$ and $b$ are not both even. Therefore, our original assumption that $\sqrt{2}$ is rational must be false, and $\sqrt{2}$ is actually irrational. This proof is often attributed to the ancient Greek mathematician Euclid, and it is still considered one of the most elegant and simple proofs of the irrationality of $\sqrt{2}$.

