# OpenReview forum: "QTIP: Quantization with Trellises and Incoherence Processing"
_NeurIPS.cc/2024/Conference — NeurIPS 2024 spotlight_

### Official Review · Reviewer_ZTfk · 2024-06-17

**Soundness:** 4
**Presentation:** 3
**Contribution:** 3
**Rating:** 6
**Confidence:** 4

**Summary:**

The paper introduces Trellis-based compression for quantization. Essentially, the method applies incoherence processing on the weight matrices, turning the approximately into Gaussians, such that the best compression method available to compression roughly Gaussian data can be applied to it. In this case, the authors choose a Trellis-based compression scheme.
They also introduce a specific Trellis setup with little overhead, for the setting where Gaussian data is the name of the game, making the method feasible in practice. According to the authors, this approach is novel.
The authors show that there is a slight benefit of this compression method over other codebooking methods, such as GPTVQ, AQLM and QUIP#, at next to no extra overhead in terms of the implementation.

As a note for the other reviewers and authors - I have no idea about trelisses, or the relevant literature. I am only knowledgeable about neural network quantization. So the veracity of some of the claims, like a novel Gaussian specific Trellis, I cannot comment on.

**Strengths:**

- The paper is very well-written, and quite easy to follow. It also feels like it's high-quality, well-researched, and the authors clearly know this field and their Trellis field very well.
- I have not seen this Trellis-based compression scheme in the deep learning literature before, so to my knowledge it is novel
- I can believe this compression scheme is better than normal VQ-based methods. Once you've turned your data into Gaussians, anything that compresses Gaussian distributed data better is likely to work.
- The paper actually implemented the idea on GPU kernels, and show that the overhead is negligible. Essentially saying the improvements are free.

**Weaknesses:**

- The results are not TOO significant. In terms of perplexity, we're talking 0.1 ppl compared to QUIP# here. And the zero-shot results are always noisy, so tough to say what's faster.
- Because the compression is likely theoretically slightly more accurate, what remains is the inference-speed of this method. Now the authors, to their credit, do address this with their GPU implementation. But, because the improvements in perplexity are so little, for fair comparison we have to be absolutely sure that the latency for the kernel is the same. Otherwise, we might as well increase the codebook size for QUIP# a little and get a better trade-off, or have one layer in a slightly higher bit-width or whatever. The Speed analysis in Table 7 does not totally satisfy my need for a thorough analysis on this. Since this is so important for the paper, I would like to see an analysis for the kernels themselves, analyze the roofline points for the band-width/compute trade-offs (does your method do more compute than QUIP#?), and have this done for more bit-widths. I believe the authors have to be more thorough on this, else I would not recommend anyone implementing the paper, and the trellis compression is relegated to being a fun curiosity instead of a practical method.
- We don't all run things on GPUs - there are devices, with hardware accelerators, CPUs etc. What does hypothetical overhead look for this method compared to simple codebook look-ups? Some hardware has actual Sillicon for codebook look-ups, or specific instructions for it like in ARM CPUs. Would your method in these cases also still be faster? Since quantization is not only a 'speed-things-up-on-gpu' topic, I would like to see a discussion on this in the paper.

Editorial Notes
5 - […] PTQ approaches have converged. I don’t think convergence has really happened, so papers do this, some don’t.
24-32 What is k? it’s undefined in the intro
160 psuedorandom -> pseudorandom

**Questions:**

Is section 24-32 entirely correct? Technically you have 2^kd vectors, but nobody ever fills out the whole space with the codebook. So it’s always a subset \subset R^{2^kd} x d you are selecting for your codebook C. So it’s not really fair to say VQ requires exponential time and space… if you keep your codebook size similar, it does not require exponential time and space. Similarly, the O(2^{kd}d ) complexity is a bit misleading. You can also say the lookup is log(|C|) depending on the size of the codebook you chose. I am not being festicious here, vector quantization methods often take a fixed set of vectors

Is the analysis is 116 sensible? Why would one ever have an unstructured k-bit codebook? There is always a way to add structure.

Table 1 - Is this done keeping the overhead of each method the same? I can always increase the dimensionality of VQ to get lower values right? This comparison should only make sense if the compute/storage overhead is the same.

How does this Trellis thing actually work? I'm trying to follow figure 2. I'm assuming the firs thing you need to store is the starting value 00. And then you follow the up-and-down errors each costing you a bit, so the red arrows should give you down, down, up, down down... So how do we end up with 0010110? Wouldn't it be 00 | 00 1 00 or 00 | 11 0 11?
Also - what happens in your code if the next value cannot be transitioned to? In your example, if 0.1 follows 0.3, you're out of luck... does it then give you the closest value 0.3?

Do you have a simple explanation for why Trellis-based coding would work better than normal codebooks? Just 'dimensionality' doesn't sit well with me, because that doesn't really explain anything concrete.


If I get more confidence from the authors on the speed of the kernels and the overhead of this method, with a discussion about non-gpu devices, especially given the very small gains they achieve - I am happy to increase my rating.

**Limitations:**

No pain points here

---

> ### Author Rebuttal · Authors · 2024-08-07
>
> ## QTIP’s significance (W1)
>
> Tables 3 and 4 show QTIP’s strengths. Table 3 indicates that QTIP *w/out fine-tuning* consistently beats VQ-based approaches *w/ fine-tuning*, showing that there is significant value in better quantizers. QTIP also succeeds where fine-tuning “fails.” Fine-tuning does not reliably improve QuIP# at 4 bits, but QTIP consistently outperforms QuIP# there. This is even more impressive since 4 bit QuIP# is almost lossless. Table 4 shows that even after fine-tuning, QTIP still consistently improves over QuIP#. At all bitrates, QTIP can reduce QuIP#’s perplexity gap by >25%, an impressive feat.
>
> ## Inference speed analysis (W2)
>
> Decoding QuIP#’s E8P takes roughly 3 instructions per weight. This is similar to QTIP’s codes. However, E8P reads 8 weights at once while QTIP reads fewer (2 for HYB, none for 3INST/1MAD). The achievable decoding speed for both methods is dependent on hardware properties (**see comments for more details**). However, QTIP’s strength lies in its flexibility. The general QTIP framework just requires efficiently generating a pseudorandom Gaussian. The paper gives 3 very different code constructions as examples, two of which don’t even require lookups. In contrast, E8P is very rigid and specially constructed. QuIP# wouldn’t work on devices with little cache, and still scales exponentially w.r.t. dimension and bitrate. For example, even a 3 bit codebook using E8P’s construction wouldn’t fit in cache on modern GPUs. QuIP# uses residual quantization to scale, but RQ is suboptimal vs. a single quantizer.
>
> ## Other hardware (e.g. ARM) (W3)
>
> Different hardware has different properties and instructions, but we believe that QTIP can be fast on a broad class of accelerators due to its flexibility. QTIP only requires generating a pseudorandom Gaussian efficiently, and can work on devices with no cache as well as devices with lookup hardware. The paper mainly targets Nvidia GPUs due to their popularity. That said, we answer this question from two perspectives:
>
> **Can look-up instructions replace TCQ?** Generally no, if you want the same quantized model quality. The look-up instructions in most architectures are limited. For example, ARMv8 NEON’s vqtbl4q_u8 intrinsic looks up 16 idxs in a 64-entry codebook. QuIP#’s E8P has 256 8D entries. We know that QTIP outperforms QuIP#, and vqtbl4q_u8 can’t even handle QuIP#. Reducing codebook size would reduce quality, so only using look-up instructions aren’t enough for QTIP’s quality.
>
> **Can look-up instructions work with TCQ?** Yes, this is essentially the HYB code. In the ARM example, we can use a 6 bit 1D codebook with HYB (Q=6, V=1). Quantizing Llama 2 7B to 2 bits with this setup and w/out fine-tuning gives 6.89 Wikitext2 perplexity – essentially the same as 3INST. We could implement this by packing the 6 bit codebook into registers and using vqtbl4q_u8.
>
> ## Editorial Notes
>
> [5] To the best of our knowledge, AQLM and QuIP# are the highest quality LLM PTQ methods currently available. Both perform vector quantization. [24-32] K is the bitrate; we will fix this.
>
> ## L24-32, exponential space and time (Q1)
>
> We’re not sure what you mean here. VQ with an unstructured codebook takes $O(2^{kd}d)$ space and time since closest vector search is NP-hard (https://cseweb.ucsd.edu/~daniele/papers/CVPP.pdf). “Exponential” refers to how the space and time complexity of VQ scale with dim and bitrate, not whether it is possible to perform VQ fast. In practice, people do use structured codebooks to reduce cost by a constant factor. This is what QuIP# uses to do 8D VQ for roughly the cost of 4D VQ (256X reduction). However, this doesn’t solve exponential scaling – it just makes VQ cheaper.
>
> ## L116, discussion on structured search (Q2)
>
> Searching over an unstructured k bit T dim codebook requires $O(2^{kT})$ time. You’re right that adding structure can make this tractable. For example, product quantization (PQ) with a group size of g makes the codebook the outer product of a g-dim codebook, reducing the runtime to $O(T2^{kg}/g)$. However, this is no longer simply VQ. TCQ is another way to add structure by making the codebook entries fixed-length walks on a trellis. The tradeoff to adding structure is increased distortion. The rate-distortion limit lower-bounds the distortion for an unstructured k bit quantizer. Table 1 shows that TCQ gets much closer to this limit than vector/scalar PQ, making it a better way to add structure.
>
> ## Table 1 (Q3)
> Table 1 computes distortion w/out considering speed. Inf-dim VQ would achieve the rate-distortion limit. 8D VQ with E8P is the limit of what we can do for fast inference on current GPUs, so the fact that QTIP can reduce the rate-distortion gap by >2/3 while using fewer resources makes it a much better quantizer.
>
> ## Figure 2 (Q4)
>
> In the bitshift trellis, the last L-kV bits of a state are the first L-kV bits of the next state, so we only need to store the kV bit “delta” for each entry in the input sequence. 0010110 represents *node indices* 00, 01, 10, 01, 11, and 10. Your coding scheme stores which edge to take, but would require knowing the graph structure to decode. The bitshift trellis bakes the structure into the numbering scheme, enabling fast decoding. Regarding transitioning to the next value, this is why we want a large trellis (large L) and randomized codebook, since that increases the probability we can transition to arbitrary values. Naive TCQ has exponential space complexity in L, so we need QTIP’s compute-based codes that dissociate L from space to get high quantization quality on today’s hardware.
>
> ## Why does TCQ work? (Q5)
>
> Mao and Gray’s RPTC paper (https://ieeexplore.ieee.org/document/5895067/) has theory on TCQ. In short, TCQ satisfies the four necessary conditions of an optimal quantizer. Intuitively, TCQ performs better because it is stateful, letting you “switch” between which size $2^{KV}$ part of the larger size $2^L$ space you are searching in at every step.

---

> ### Author Response · Authors · 2024-08-07
> **More discussion on QTIP inference speed analysis (comment for rebuttal to W2)**
>
> In general, reducing the size of the uncompressed codebook reduces the cost of lookup. QuIP# is forced to use a battery of bit manipulation operations (amortized to 3 per weight) to compress its 65536-entry V=8 codebook down to a 256*32-bit LUT. Due to the trellis-based quantization algorithm, QTIP HYB requires a much smaller uncompressed codebook (1024 entries at V=2) to achieve better accuracy. We decompress this from a 512 * 32-bit LUT in only 0.5 integer ALU instructions per weight, versus QuIP# which requires 3. QTIP HYB has an additional cost of ~1.5 integer ALU ops to unpack the trellis and calculate the codebook index, which comes out to a total of 2 ALU ops per loaded weight. At 2 bits, this is 8 ALU ops per loaded byte. On the flip side, QTIP at V=2 requires 4x more codebook lookups than QuIP# at V=8. The effect of these differences on inference speed depends on the details of the specific target microarchitecture.
>
> Essentially,
> - QTIP reduces the necessary uncompressed codebook size, which reduces the necessary runtime resources (smaller LUT hardware or fewer ALU ops: 0.5 ALU ops for HYB vs 3 for QuIP# E8P).
> - QTIP HYB requires 1.5 ALU ops per weight to unpack and map trellis entries to codebook indices.
> - QTIP HYB requires more lookups per weight than QuIP#. This is fine on modern GPUs, which have fast banked shared memory. On other architectures, performance will depend on the SIMD shuffle or L1 cache throughput. However, QTIP can be done without any lookups at all (e.g. 3INST and 1MAD) while QuIP# cannot.
> - The decompress overhead is essentially constant for all bit widths for QTIP. VQ costs more as bit width increases.
> - QTIP 3INST and 1MAD require a fixed number of ALU instructions to decode, and do not use a LUT, reducing cache considerations.
>
> where “ALU ops” here refers to INT32 ALU ops.
>
> **Openreview is not letting me make a comment to the rebuttal, so you are going to see this before the rebuttal. Please read the rebuttal first before reading this.**

---

> ### Comment · Reviewer_ZTfk · 2024-08-08
> **Further questions**
>
> Some more questions and comments:
>
> Thanks for your explanation on the Trellis decoding - I believe it would also be wise to add this explanation to the paper, as as a layman in the information theoretical source coding field, I did not get it at first.
>
> I am still unclear on what the Lookup-free computed codes actually do. As I understand it, you can take any word, run it through your 1MAD or 3INST algorithm, and out comes the number you decode right? Do they still have anything to do with the trellis coding that you mentioned? Given a weight tensor, how do you actually find which words encode this value the best?
>
> On the discussion of the runtime and it’s exponential time - I believe I understand what you are saying, but I also think it’s a bit unfair. Your introduction essentially reads: There’s these VQ methods, and they are exponential in the dimensionality, our method achieves better scaling in dimensionality, so it’s better…our problem sizes are fixed, we’re compressing a fixed-size network. These compression methods are never used in a method where the exponential scaling in dimensions actually matters. The authors of the other papers find an optimal trade-off in dimensions/num_clusters->codebook size and performance. This is not like a travelling salesman problem where your O() notation actually matters if you want to scale up your problem to larger sizes.  The dimensionality is a choice, and the scaling being exponential is not the actual fact that matters. Rather your question is: At the same codebook sizes/decoding time, what is the best packing I can achieve. You claim ‘VQ requires exponential […]  which limits its practicality', which I think is too strong. These methods are very practical because they are applied on lower dimensions, and the codebook size is explicitly limited. Given that this argument is so important for your paper, I would take some care :)
>
> For some of the comments I made, I was also secretely asking that they would be addressed in a new revision of the paper, any thoughts on that? ;)
>
> Editorial Note: Even if AQLM and QUIP# are the highest quality LLM PTQ methods currently known to you, it does not mean the field has converged. I don’t believe any model efficiency committee with prominent researchers created a consensus of what is best, and concluded that there is nothing better out there. If anything, it's highly dependent on what's available on the hardware as well.

---

> > ### Author Response · Authors · 2024-08-09
> > **Responses to Further Questions**
> >
> > > Thanks for your explanation on the Trellis decoding ...
> >
> > We will add it to the paper.
> >
> > > I am still unclear on what the Lookup-free computed codes ...
> >
> > These codes take a L bit word and return a FP16 number (or vector for HYB). For a uniform distribution of L bit inputs, the outputs of these codes are approximately i.i.d Gaussian. These codes are used to generate the codebook during trellis quantization. Recall in trellis quantization that the trellis has 2^L nodes/states, each with 2^{kV} directed edges to other states where k is the bitrate and V is the number of weights we quantize in a single step. Each state has a V-dimensional vector assigned to it, and a length-T input sequence is quantized to a length (T/V)-1 walk on this graph. The reconstructed sequence is the concatenation of the state values on this walk.
> >
> > For example, in Figure 2 in the paper, the walk consists of nodes (00, 01, 10, 01, 11, 10), so the reconstruction is (0.5, 0.1, 0.8, 0.1, 0.3, 0.8), since these are the values that correspond to those nodes. In QTIP, the codes are used to generate the value of node i by running the algorithm on i. Since there are 2^L nodes, i spans 0 to 2^L-1, so the codebook consists of all possible outputs of a code. This means that the codebook is approximately i.i.d Gaussian, which is what we want for quantizing a i.i.d. Gaussian source.
> >
> > To actually quantize a sequence of T weights, we can run the Viterbi algorithm on this graph. The Viterbi algorithm performs dynamic programming by storing the minimum distortion achievable at step t, 0 <= t <= T/V - 1. Section 2.3 in the paper has more details, but the algorithm itself is well documented (https://ieeexplore.ieee.org/document/1450960). In our experiments, we split the weight matrix into 16x16 tiles and quantized each one as a length 256 sequence using BlockLDLQ from QuIP#. This is just one way of applying QTIP, and you could alternatively quantize the entire matrix as a single sequence if the way you quantized (e.g. LDLQ, direct optimization a la AQLM, etc.) was compatible with that.
> >
> > > On the discussion of the runtime and it’s exponential time ...
> >
> > Can you explain why exponential scaling doesn’t matter? As you noted above, VQ is practical when applied at lower dimensions like 4 and 8 because the codebook can be kept small. We know from Table 1 and the QuIP# paper that going from 4D VQ to 8D VQ significantly improves quality, so going past 8 dimensions would further improve quality. However, 8 is the highest dimension current VQ methods can achieve while still maintaining fast inference because higher dimension codebooks would not fit in cache. This means that we are indeed limited by the size of the codebook and VQ’s space complexity; we cannot arbitrarily improve VQ’s quality by increasing dimension while still maintaining inference speed. In contrast, TCQ doesn’t suffer from these problems. We can do TCQ without any stored codebooks at all, and decoding is linear in the bitrate.
> >
> > As a side note, I don’t think we fundamentally disagree about VQ’s problems. Perhaps “practicality” is the wrong word here and something like “scalability” would be better. We’re open to using other words if you’d like to suggest one.
> >
> > > For some of the comments I made ...
> >
> > We plan on adding revisions from discussions with all reviewers after the review period.
> >
> > > Editorial Note: Even if AQLM and QUIP# are ...
> >
> > We are also open to using a different word than “converged,” if you’d like to suggest one.

---

> > > ### Comment · Reviewer_ZTfk · 2024-08-09
> > > **Further comments**
> > >
> > > Thanks for the explanation on how the codes are used. I understand it now. Essentially, you are trading off compute for memory this way. Instead of storing a codebook, there is a procedure for generating the codebook on the fly. When you decode, you run it through the function and you get e.g. 11 -> 0.332 which is the decoded value. And I’m assuming the codebooks would otherwise be much larger than the programming code to do this on-the-fly generation.
> > >
> > > I guess a final question for me to understand why this method could be better theoretically: Why would these randomly generated codes be better than something that is not randomly generated? E.g., another value generating function could just be something that maps bits to a floating-point number, arithmetically: e.g. 8 bits could map to something with 1 sign, 5 mantissa and 2 exponent bits in FP. Something we also know approximates a N~[0, 1] Gaussian well. For e.g. 4 bits you can probably find the optimal mapping of 4 bits to whatever mantissa/exponent bit configuration is the most accurate for matching a Gaussian as well. This can be done arithmetically too without an explicit codebook. What is the benefit over something that’s seemingly randomly generated over something that has such structure? One thing that worries me is that for a low number of bits, your 'approximately Gaussian' doesn't really hold very strongly, and there will be quite some bias in the number format you choose.
> > > I guess, in the end all these codes are doing are defining a different (random) way to put representable points on the number line. Could you plot for your experiments where the representable values finally lie on your number-line?
> > >
> > > On the exponential scaling. I don’t believe the exponentially of the codebook sizes in the dimensions directly matters, since the problem is rather: Given a certain accuracy, what is the most efficiency I can get. Or in codebook terms, given a codebook size (e.g. something that fits in L1 cache), what is the best accuracy I can get. This is very different from combinatorial optimization problems, where you might have a problem with trying a TSP solver on a small graph, and then when you scale it to industrial sizes the algorithms don’t work and they break because it’s too expensive. This is not the case here in this setting, which has mostly fixed constraints. Dimensionality, or scalability with respect to dimensionality, is not the goal by itself here and it’s solely subject to the accuracy/performance trade-off. I believe your story would be easier to comprehend if that was the focus, not the fact that your algorithm somehow scales better in big-O notation.
> > >
> > > It would be great if you could be a little bit more explicit with what unclarities/comments you agree with, and commit to adding to your paper, and what not. Some things that are unclear to me, like e.g. how are the look-up free codes used in the algorithm, I’m assuming would be unclear to others as well. It would be great to see if you could commit to making these things clearer in the paper for folks that have not read up on Trellis-based encoding and can infer what you are doing.
> > >
> > > Converged -> How about "Recent state-of-the-art 4 PTQ approaches have utilized using vector quantization (VQ)”

---

> ### Author Response · Authors · 2024-08-10
> **Responses to Further Comments**
>
> > I guess a final question for me to understand ...
>
> To clarify, these computed codes are not random, they just do a good job of decorrelating the state ID from the computed value. Like your example FP8 casting code, our codes are deterministic in their outputs. The reason why we want a code that appears random is because under the bitshift trellis, random codes are asymptotically optimal as L increases (see the RPTC paper for more details https://ee.stanford.edu/~gray/trellis.pdf). Table 1 in the RPTC paper shows that an optimal code with a small L has higher distortion than a random code with a large L. This means that we should always increase L as much as possible, and our compute-based codes let us do that without impacting decoding speed.
>
> > One thing that worries me is that for a low number of bits, your 'approximately Gaussian' doesn't really hold very strongly, and there will be quite some bias in the number format you choose.
>
> In trellis coding, the codebook assigns values to the *states* (2^L), so the number of representable codebook values is independent of the bitrate K. These codes are approximately Gaussian for large enough L ($\gtrapprox$ 10), and our experiments use L=16. Figure 3 shows the set of representable neighboring values in a bitshift trellis for various codebooks when L = 16. Indeed, 1MAD and 3INST give very similar coverage as a random Gaussian. Figure 3 also shows why we want a “randomized” codebook. Since neighboring states in the bitshift trellis share L-KV bits, a poorly designed computed code will have strong correlations that result in unrepresentable neighboring values. The leftmost plot in Figure 3 shows such a code, where large portions of the state space are unrepresentable with the bitshift trellis. A code that does direct casting to FP8 (what you suggested above) would have extreme correlations since the mantissa bits of a state become the exponent and sign bits of the next state. Openreview isn’t letting me send an image of the correlation plot, but you can generate a plot with the Python script below to verify for yourself. Section 3.1 in the paper also has more details on random codes.
> ```
> import torch
> import matplotlib.pyplot as plt
>
> L = 8
> # bitrate, adjust accordingly. V =1 for plotting purposes.
> K = 4
>
> states = torch.arange(1 << (L+K), dtype=torch.int32)
>
> left = (states >> K).to(torch.uint8)
> right = (states & ((1 << L) - 1)).to(torch.uint8)
>
> lval = left.view(torch.float8_e5m2).float()
> rval = right.view(torch.float8_e5m2).float()
>
> plt.scatter(lval, rval)
> plt.show()
>
> ```
>
> > On the exponential scaling ...
>
> The focus of the paper is indeed to show that QTIP achieves a better quality/speed tradeoff over existing quantization methods. Our empirical results show that QTIP outperforms VQ-based methods such as QuIP# and AQLM while offering the same fast inference as QuIP#. The section about VQ’s exponential scaling was to explain why we can’t increase the VQ dimension to improve quality while preserving fast inference. We know from information theory that the distortion of a vector quantizer is lower bounded by a strictly decreasing function of the dimension (http://vkostina.caltech.edu/pdfs/2012KostinaVerdu-lossycomp.pdf). This means that for optimal codebooks and a fixed bitrate, the only way to improve VQ’s quality is by increasing the dimension. TCQ lets us achieve lower distortion than the best “fast inference” VQ, which translates into higher quality quantized LLMs. QTIP’s contribution is enabling fast decoding with TCQ so we can quantize LLMs with it while supporting fast inference.
>
> > It would be great if you could be a little bit more explicit ...
>
> Most of our responses are based off of information already in the paper. We will make this information more clear in an updated manuscript. We will also add new information from the rebuttal/discussion period, such as those on hardware requirements for QTIP's kernels.

---

> > ### Comment · Reviewer_ZTfk · 2024-08-12
> > **Update on score**
> >
> > After finally understanding the paper properly, I believe I can now say that I think the QTIP method is solid and worthwhile considering as a method to encode and decode weights for LLM inference. I think the approach is novel enough on top of existing codebook approaches. Although I think the result improvements are still minor, and thus do not warrant a best-paper award, I believe any smidgeon of improvement counts. If we were to reject a paper just because the improvement is small, we'd have a big problem in our field of efficient deep learning. Especially since the benefits would be essentially free, if a suitable implementation exists.
> >
> > I do however still have two import notes:
> > - It took me quite a while to understand the method - and with me likely the average reader that's not inundated in the field of Trellis codes. I would strongly encourage the authors to more clearly explain their method in the paper, so new readers will not have to resort to Openreview or send you an e-mail to understand what is going on.
> > - I am also a bit partial to reviewer F569's question on showing the benefit of this code versus other codes for more examples. Incoherence processing only approximately and probabilistically makes your distributions Gaussian - I've seen that in practice outliers can still occur, and we're left guessing now what happens with this method in such cases. A more extensive expose of this method on more than a few distributions that occur in the wild would do it good.
> >
> > That said, the positives outweigh the negatives and I've increased by score to a weak accept.

---

> ### Author Response · Authors · 2024-08-13
> **Thanks**
>
> Thank you for your review, we will be sure to clarify how trellis coding works in an updated manuscript. We uploaded a response to F569's latest questions, which you may find useful.

---

### Official Review · Reviewer_F569 · 2024-07-08

**Soundness:** 3
**Presentation:** 3
**Contribution:** 3
**Rating:** 5
**Confidence:** 4

**Summary:**

This paper proposes QTIP, a new method for efficient post training quantization of LLMs. QTIP is a vector quantization method inspired from Quip# [1], but with no limitation in the dimension of the codebook. The work’s fundamental contributions are: 1) Propose to adapt trellis quantization to the compression of LLM pre-trained matrices. 2) Design efficient trellis and codebook in dimension (or 'sequence length') $=256$. 3) Integrate the trellis quantization into the Quip# [1] optimization process to fine-tune the pre-trained weights. The numerical experimental results of the proposed method outperform AQLM [2] and Quip# [1] on the C4 and Wikitext2 datasets.

---

[1] Albert Tseng, Jerry Chee, Qingyao Sun, Volodymyr Kuleshov, and Christopher De Sa. Quip#: Even better llm quantization with hadamard incoherence and lattice codebooks, 2024.

[2] Vage Egiazarian, Andrei Panferov, Denis Kuznedelev, Elias Frantar, Artem Babenko, and Dan Alistarh. Extreme compression of large language models via additive quantization, 2024.

**Strengths:**

1) The authors proposed an interesting description of the limitations (in terms of codebook dimension) related to standard vector quantization, and a sound introduction to trellis quantization.
2) This paper presents two innovative lookup-free code for (non-fine-tunable) trellis weight quantization: '1mad' and '3inst'. These methods obtain sota performances with respect to other non-fine-tuned approaches, and are efficiently implemented without the need to store the codebooks.
3) The author also propose an hybrid (fine-tunable) trellis weight quantization: 'hyb'. Though 'hyb' requires to store the codebook, it allows one to fine-tune the codewords, and thus enable QTIP to compete with fine-tuned sota approaches [1,2].

---

[1] Albert Tseng, Jerry Chee, Qingyao Sun, Volodymyr Kuleshov, and Christopher De Sa. Quip#: Even better llm quantization with hadamard incoherence and lattice codebooks, 2024.

[2] Vage Egiazarian, Andrei Panferov, Denis Kuznedelev, Elias Frantar, Artem Babenko, and Dan Alistarh. Extreme compression of large language models via additive quantization, 2024.

**Weaknesses:**

1) QTIP is mainly based on Quip# [1]: 'we use QTIP as a quantizer in QuIP#’s BlockLDLQ' (line.224). The code is not provided, but the authors explain that the whole quantization and fine-tuning process is based on Quip# [1]. Hence the contribution is more to the field of signal quantization in general, and LLM compression is an application.
2) QTIP natively reuses the incoherence processing from [1]. But this pre-processing step may erase the native correlation patterns that pre-trained LLM matrices bear [2,3]. No discussion about this process (and the blockLDLQ) is provided.
3) QTIP (with fine-tuning) update the pre-trained weights 'in a blockwise fashion' (line.247). This step is computationally intensive regarding the size of the matrices (and the optimizer states related to it), and it may impact the efficiency of the quantization process.
4) Very few details are given regarding the experiments design (which calibration dataset is used for fine-tuning 'hyb' codewords?), what may affect the results reproducibility (in addition the code is not publicly available).

---

[1] Albert Tseng, Jerry Chee, Qingyao Sun, Volodymyr Kuleshov, and Christopher De Sa. Quip#: Even better llm quantization with hadamard incoherence and lattice codebooks, 2024.

[2] Kim, S., Hooper, C., Gholami, A., Dong, Z., Li, X., Shen, S., Mahoney, M. W., and Keutzer, K. (2023). Squeezellm: Dense-and-sparse quantization. arXiv preprint arXiv:2306.07629.

[3] Guo, H., Greengard, P., Xing, E., and Kim, Y. (2023). Lq-lora: Low-rank plus quantized matrix decomposition for efficient language model finetuning. In The Twelfth International Conference on Learning Representations.

**Questions:**

1) Table.1 shows the superiority of QTIP over scalar quantization and Quip# [1]. How does it translate to the case of 'real' pre-trained LLM matrices? Can you show some results similar to [2], to identify which matrix presents the lowest quantization error ? This would enable us to decouple the gains provided by QTIP, and understand better which part (the quantization process itself, or the fine-tuning machinery) is key for sota performances.
2) Is 'hyb' codebook unique? Or do you have a different (fine-tunable) codebook for each LLM block?
3) Do you think QTIP fine-tuning step is (computationally) competitive with respect to (low-rank) adapters methods such as LoRA [3] and LQ-LoRA [2] ? Or QTIP is only tailored for LLM compression ?


---

[1] Albert Tseng, Jerry Chee, Qingyao Sun, Volodymyr Kuleshov, and Christopher De Sa. Quip#: Even better llm quantization with hadamard incoherence and lattice codebooks, 2024.

[2] Guo, H., Greengard, P., Xing, E., and Kim, Y. (2023). Lq-lora: Low-rank plus quantized matrix decomposition for efficient language model finetuning. In The Twelfth International Conference on Learning Representations.

[3] Hu, E. J., Wallis, P., Allen-Zhu, Z., Li, Y., Wang, S., Wang, L., Chen, W., et al. (2021). Lora: Low-rank adaptation of large language models. In International Conference on Learning Representations.

**Limitations:**

A comparison of inference speedups is provided in section 4.3.

---

> ### Author Rebuttal · Authors · 2024-08-07
>
> ## QTIP's contribution (W1)
> The focus of QTIP is on *what to quantize with* (e.g. VQ, TCQ), and not *how to quantize* (e.g. GPTQ, fine-tuning). Choosing a good quantizer is hard since LLM weight matrices have outliers and small-batch LLM inference is memory bound, necessitating fast decoding. While the field of signal processing has known for a while that TCQ achieves lower distortion than VQ, there is little to no existing work on designing trellis quantizers for fast parallel decoding. This is the “missing piece of the puzzle” needed to make TCQ work for LLM compression, which QTIP addresses.
>
> Our experiments used QuIP#’s BlockLDLQ framework for two main reasons. First, QuIP# is a state-of-the-art vector quantization framework that supports fast inference, making it a good baseline to show the benefits of TCQ. Second, LDLQ variants are optimal among adaptive rounding methods that use linear feedback from the Hessian. By using BlockLDLQ, we can be reasonably certain that the empirical improvements are from TCQ and not spurious interactions between the quantizer and a subpar rounding algorithm. Finally, we note that QTIP could be used as a drop-in replacement for VQ in other rounding algorithms. QTIP’s orthogonality to the actual rounding method further broadens QTIP’s practicality.
>
> ## Incoherence Processing (W2)
>
> Both QuIP and QuIP# detail how incoherence processing affects the Hessian and how we can use incoherence to bound the proxy error tr(WHW.T). We refer the reviewer to QuIP and QuIP# to see how incoherence processing can benefit quantization, even when using Hessian information.
>
> QTIP uses the random Hadamard transform (RHT) to perform incoherence processing. The RHT is fully invertible up to numerical error, meaning that the Hessian matrices do not lose information from incoherence processing bar catastrophic imprecision. We ran the quantization algorithm in fp64 and did not observe differences vs. fp32, so numerical error is not a dominating factor in the quantization step.
>
> ## Cost of Fine-Tuning (W3)
>
> QTIP’s focus is on the quantizer itself and not fine-tuning. Whether the end user chooses to fine-tune or not is a “deployment choice,” and in either case QTIP offers improvements over SOTA vector quantization methods. Table 3 shows that *even without any fine-tuning*, QTIP usually outperforms SOTA vector quantization based methods *that use fine-tuning*, meaning that fine-tuning isn't strictly necessary to make QTIP SOTA. Table 4 shows that QTIP still gives significant improvements after fine-tuning. In fact, QTIP holds up where fine-tuning “fails.” QTIP almost halves the 4 bit perplexity gap vs. QuIP#, while fine-tuning has little effect at 4 bits. Finally, since QTIP is orthogonal to fine-tuning, we expect that new fine-tuning algorithms will compose with QTIP to produce even better quantized models.
>
> ## Implementation Details (W4)
> Most of these details, including datasets, are available in the Appendix. We will add additional information in an updated manuscript and the code will be made available at a later date.
>
> ## Quantization Error (Q1)
> Table 1 shows the distortion for an i.i.d. Gaussian sample. If we re-run Table 1 with actual weight matrices and incoherence processing, then we get very similar results since incoherence processing produces approximately i.i.d. Gaussian weights.
>
> |           Source           |  1MAD MSE | 3INST MSE |  HYB MSE |
> |:--------------------------:|:-----:|:-----:|:-----:|
> | i.i.d. Gaussian (Table 1)  | 0.069 | 0.069 | 0.071 |
> |    Llama 2 7B 0_v + IncP   | 0.069 | 0.069 | 0.070 |
>
> However, LLM quantization algorithms usually minimize the expected activation error (proxy error, Eq 1) instead of MSE (Table 1). Using a calibration set of 25 million tokens from RedPajama, scalar quantization achieves a relative error of 0.073, VQ (QuIP# E8P) 0.060, and TCQ (QTIP 3INST) 0.045, when quantizing the 10_down layer of Llama 2 70B to 2 bits without fine-tuning. Like in Table 1, QTIP reduces the proxy error over SQ and VQ.
>
> ## HYB Uniqueness (Q2)
> In our experiments, we use a different codebook per linear layer. Since this codebook is very small (1K entries) relative to the matrix size (>>1M entries), it adds < 0.01 bits per weight. If a per-layer codebook is not possible, a shared codebook should also work fine. 1MAD and 3INST use the same codebook per layer and also achieve SOTA performance. This is due to incoherence processing producing approximately i.i.d Gaussian weights, so the codebooks are all essentially Gaussian.
>
> ## LLM Quantization Fine-Tuning vs. LoRA-Style Methods (Q3)
>
> To clarify, QTIP’s focus is on the quantizer, not the rounding or fine-tuning algorithm. Table 4 used QuIP#’s tune-as-you-go fine-tuning method. This method, and other similar ones like AQLM’s fine-tuning algorithm, tune the quantized model to recover the original model. In contrast, LoRA and LQ-LoRA are aimed at fine-tuning to new downstream tasks. These are two fundamentally different problems, so it is difficult to compare methods across problem areas. It may be possible to adapt methods to solve both problems, but that is beyond the scope of this work.

---

> > ### Comment · Reviewer_F569 · 2024-08-12
> >
> > Thank you for your answers. However, my Q.1 remains, 'how to decouple the gains provided by QTIP, and understand better which part (the quantization process itself, or the fine-tuning machinery) is key for sota performances?'. Your rebuttal provides a comparison of a single LLaMA layer and iid Gaussian vectors, but I believe this comparison is not informative regarding Q.1. A fair comparison would detail the expected activation error (proxy error, Eq 1) or the MSE for QTIP **and** other methods (e.g. LQLoRA, Quip#, AQLM, etc.) for all layers in a LLM (I believe some layers would suffer more from the incoherence processing step; see for e.g. early layers in Fig.1 in [1]). I will keep my score.
> >
> >
> >
> > [1] Guo, H., Greengard, P., Xing, E., and Kim, Y. (2023). Lq-lora: Low-rank plus quantized matrix decomposition for efficient language model finetuning. In The Twelfth International Conference on Learning Representations.

---

> > > ### Author Response · Authors · 2024-08-13
> > > **Response to comment part 1**
> > >
> > > Splitting into multiple parts due to the character limit
> > >
> > > > However, my Q.1 remains, 'how to decouple the gains provided by QTIP, and understand better which part (the quantization process itself, or the fine-tuning machinery) is key for sota performances?'.
> > >
> > > The main difference between our QTIP experiments and QuIP# is the quantizer. The QTIP experiments use our fast trellis quantizer and QuIP# uses a vector quantizer. Therefore, the difference between QTIP and QuIP# is the difference from using a better quantizer. Our QTIP experiments and QuIP# both use the BlockLDLQ rounding algorithm and the same fine-tuning algorithm.
> > >
> > > We are not claiming that trellis coding alone is sufficient for state of the art quantization performance. Doing so would ignore the vast model quantization literature that shows the efficacy of algorithms such as adaptive rounding and fine-tuning. Rather, all we are saying is that trellis coding's gains over vector quantization translate to LLM quantization, *even after adaptive rounding and fine-tuning*. This means that although adaptive rounding and fine-tuning are important to achieve strong performance, there is still a significant benefit to using a better quantizer. QTIP lets us achieve fast decoding with trellis coding, making trellis coding practical for LLM quantization.
> > >
> > > > Your rebuttal provides a comparison of a single LLaMA layer and iid Gaussian vectors, but I believe this comparison is not informative regarding Q.1.
> > >
> > > This table was in response to "How does it translate to the case of 'real' pre-trained LLM matrices? Can you show some results similar to [2], to identify which matrix presents the lowest quantization error ?" Our understanding is that Figure 1 in the LQ-LoRA paper measures the RMSE (sqrt distortion) of the original weight matrix. The table in the rebuttal does this as well and shows that after incoherence processing, "real pre-trained LLM matrices" have the same quantization distortion as an i.i.d Gaussian with QTIP. This is significant because, to the best of our knowledge, trellis coding achieves the lowest empirical distortion on memoryless Gaussian sources (https://ee.stanford.edu/~gray/trellis.pdf). Having this result translate to LLM matrices means that QTIP is indeed a better quantizer than VQ, even on actual pre-trained matrices.

---

> ### Author Response · Authors · 2024-08-13
> **Response to comment part 2**
>
> > A fair comparison would detail the expected activation error ...
>
> The rebuttal does have this information for the 10_down layer of Llama 2 70B (see the paragraph after the table). QTIP achieves a lower proxy error than QuIP (scalar quantization) and QuIP# (vector quantization). We are not familiar with the AQLM codebase so we did not attempt to measure its proxy error. Due to time, we won't be able to get proxy error measurements for all the layers before the end of the discussion period. However, below are proxy errors for layers 39 and 79 of Llama 2 70B for scalar quantization, vector quantization (QuIP#), and trellis quantization (QTIP) when quantized to 2 bits. As expected, QTIP achieves the lowest proxy error in all cases.
>
> **Note: there was a bug in the scalar quantization proxy error code in the first version of this response. We have updated the table with the correct results. The SQ proxy error in the original rebuttal was not affected by this bug.**
>
> |  Layer  | 2 Bit Scalar Quantization Proxy Error | 2 Bit Vector Quantization (QuIP#) Proxy Error | 2 Bit Trellis Quantization (QTIP) Proxy Error |
> |:-------:|:-------------------------------------:|:---------------------------------------------:|:---------------------------------------------:|
> |   39 q  |                 0.0105                |                     0.0088                    |                   **0.0065**                  |
> |   39 k  |                 0.0091                |                     0.0074                    |                   **0.0056**                  |
> |   39 v  |                 0.0567                |                     0.0458                    |                   **0.0341**                  |
> |   39 o  |                 0.0372                |                     0.0318                    |                   **0.0246**                  |
> |  39 up  |                 0.0626                |                     0.0531                    |                   **0.0407**                  |
> | 39 gate |                 0.0435                |                     0.0367                    |                   **0.0282**                  |
> | 39 down |                 0.0680                |                     0.0582                    |                   **0.0441**                  |
> |   79 q  |                 0.0064                |                     0.0054                    |                   **0.0041**                  |
> |   79 k  |                 0.0049                |                     0.0041                    |                   **0.0031**                  |
> |   70 v  |                 0.0453                |                     0.0387                    |                   **0.0291**                  |
> |   79 o  |                 0.0069                |                     0.0053                    |                   **0.0041**                  |
> |  79 up  |                 0.0116                |                     0.0099                    |                   **0.0075**                  |
> | 79 gate |                 0.0116                |                     0.0099                    |                   **0.0075**                  |
> | 79 down |                 0.0012                |                     0.0010                    |                   **0.0007**                  |
>
> Regarding LQ-LoRA, our understanding is that LQ-LoRA focuses on obtaining a better quantized + low rank initialization for the final goal of downstream fine-tuning to new tasks. Post training quantization methods like QuIP#, AQLM, and QTIP focus on finding a quantized model that is *as close to the original model as possible*, which is a fundamentally different problem. LQ-LoRA mainly presents results on downstream fine-tuning, where they can get lower perplexity on C4 than even the original unquantized model. This is impressive, but should also make it obvious that LQ-LoRA is solving a different problem than PTQ.
>
> > I believe some layers would suffer more from the incoherence processing step; see for e.g. early layers in Fig.1 in [1]
>
> We're not sure what you mean by "suffer more" from incoherence processing. Incoherence processing is a fully invertible transformation up to numerical error. This means that incoherence processing does not result in information loss unless there is catastrophic numerical imprecision. As mentioned in the rebuttal, we did not observe differences from running our experiments in FP32 vs. FP64, so numerical imprecision is not an issue here. If you are questioning the efficacy of incoherence processing in the quantization process, we recommend taking a look at QuIP and QuIP#. Both papers have theory showing that the proxy error can be bounded by the incoherence $\mu$ of the weight matrix when using LDLQ variants. The bound gets tighter as $\mu$ gets smaller, which is what incoherence processing does.

---

### Official Review · Reviewer_1CRX · 2024-07-11

**Soundness:** 4
**Presentation:** 4
**Contribution:** 4
**Rating:** 7
**Confidence:** 4

**Summary:**

This paper introduces trellis coded quantization (TCQ) into large language model (LLM) quantization, achieving ultra-high-dimensional quantization with less inference burden compared to traditional vector quantization (VQ) methods. The main innovations are a hardware-efficient "bitshift" trellis structure and fast compute-based Gaussian codes, enabling high-quality quantization and fast inference.

**Strengths:**

1. The motivation to address the drawbacks of VQ is clear and intuitive.
2. The proposed "bitshift trellis" and "compute-based random Gaussian codes" are innovative and improve the inference efficiency of TCQ.
3. The performance is impressive, significantly enhancing 2-bit performance even without fine-tuning and optimizing quantization bits to 3-bit.
4. The method significantly improves inference speed.

**Weaknesses:**

This is a solid paper introducing a novel and robust quantization format. Inference efficiency is crucial for new quantization formats, but this paper only presents the inference speed on the RTX 4090 with 2-bit quantization. Providing inference speeds for more bits (e.g., 3-bit, 4-bit) and more devices (e.g., RTX 3090, A100) could enhance the paper further.

It is possible that existing quantization computation kernels achieve speedup on the RTX 4090 but not on the A100. Such phenomena are normal, and adapting a new kernel to different devices is challenging. However, I encourage the authors to report inference speeds on various devices. Even if results are not favorable, an in-depth analysis with potential solutions would provide a more comprehensive understanding without affecting the paper's contribution.

**Questions:**

Please see the weaknesses for details.

**Limitations:**

The paper lacks inference speed testing on more bit levels and additional devices.

---

> ### Author Rebuttal · Authors · 2024-08-07
>
> ## Updated Inference Speeds on More Devices and More Bitrates
>
> Below are throughput numbers for decoding 1024 tokens on the RTX 3090, RTX A6000 Ampere, and RTX 6000 Ada, averaged over 8 runs. The kernels were not re-tuned for each device, so they could be made faster. These numbers were run using an updated inference speed script from the QuIP# codebase that is more accurate than the numbers in the paper. In all cases where the FP16 model fits on the device, QTIP is faster than FP16.
>
> |     GPU Model    | Model  | 2-bit tok/s | 3-bit tok/s | 4-bit tok/s | FP16 tok/s|
> |:----------------:|:------:|:-----------:|:-----------:|:-----------:|:----:|
> |     RTX 3090     |  2-7b  |     127     |     119     |     109     | 52.5 |
> |     RTX 3090     |  2-70b |     15.3    |     OOM     |     OOM     |  OOM |
> | RTX A6000 Ampere |  2-7b  |     116     |     106     |      95     | 43.5 |
> | RTX A6000 Ampere |  2-70b |     15.0    |     13.1    |     11.7    |  OOM |
> |   RTX 6000 Ada   |  2-7b  |     188     |     161     |     140     | 55.9 |
> |   RTX 6000 Ada   |  2-70b |     23.5    |     19.1    |     16.3    |  OOM |

---

> > ### Comment · Reviewer_ZTfk · 2024-08-08
> > **FP8?**
> >
> > Wouldn't a ~2x speed-up of e.g. 4-bit tok/s compared to FP16 tok/s get negated when the FP8 format is used?
> > Also, wouldn't the fairer comparison be more of what happens when you use 4-bit native kernels versus 4-bit kernels with the trellis coding?

---

> > > ### Author Response · Authors · 2024-08-09
> > > **Hardware Supported Datatypes and Other Quantizers (re ZTfk)**
> > >
> > > >Wouldn't a ~2x speed-up of e.g. 4-bit tok/s compared to FP16 tok/s get negated when the FP8 format is used? Also, wouldn't the fairer comparison be more of what happens when you use 4-bit native kernels versus 4-bit kernels with the trellis coding?
> > >
> > > Not really. QTIP 4 bit is actually closer to 3X faster than FP16 on Ada/Hopper, which is what you’d need for hardware FP8 support. Of the 3 GPUs we tested on, only the RTX 6000 Ada has hardware support for FP8, so you wouldn’t get any compute speedup from FP8 on the 3090 and A6000 Ampere. QTIP could also be made faster by fusing some of the Hadamard transforms like QuaRot does “for free”. We can test this by fusing the same RHTs as QuaRot’s. Here, we get 151 tok/s on the A6000 Ada for 7B 4 Bit.
> > >
> > > Quantizing to 8 bits also won’t give a 2X inference speedup due to overhead from other parts of the model. We can emulate inference with an 8 bit model by simply dividing half the matrix dimensions by 2 in a FP16 model, which conveniently means doing half the compute and reading half the memory. The alternative would be using the TransformerEngine FP8 GEMM kernel, which is complicated to integrate. On an A6000 Ada, this gives 98.4 tok/s for Llama 2 7B, which is only 1.77x the speed of FP16 and 35% slower than 4 bit QTIP with fused RHTs.
> > >
> > > Another caveat with using hardware supported datatypes like FP8 and INT4 is needing to quantize *both the activations and weights*, since we don’t have hardware support for mixed precision GEMMs. Quantizing the activations means sacrificing quality, and existing methods give worse performance/speed tradeoffs than QTIP. For example, to the best of our knowledge, QuaRot and SpinQuant are the two best performing W4A4 methods, and they both perform significantly worse than 4 bit QTIP (QuaRot 6.10 ppl, SpinQuant 5.9 ppl, QTIP 5.52 ppl, FP16 5.47 ppl, Wikitext 2, ctx. 2048). SmoothQuant’s W8A8 setup does have similar quality as 4 bit QTIP, but here again the speedup will be less than 4 bit QTIP in memory bound scenarios (see above).
> > >
> > > However, we don’t actually need hardware support for smaller datatypes to do memory bound inference. As long as we can write a kernel to do decompression during the matrix multiplication, then we can use any quantizer. This is what QTIP and existing weight-only quantization methods like QuIP# and AQLM do. To test speed here, we used Microsoft’s BitBLAS library as a drop-in matmul in torch’s Linear layer. BitBLAS contains highly optimized kernels to do FP16 x dtype matmuls, where dtype is an IEEE dtype. Since these datatypes are much easier to dequantize than QTIP’s codes, these kernels should be about as fast as possible for a weight-only quantization method. BitBLAS FP16 x INT4 gets 150 tok/s on the A6000 Ada, which is essentially the same as QTIP’s 151 tok/s. However, QTIP is still doing some RHTs, so QTIP’s 4 bit kernel is actually faster than BitBLAS’s 4 bit kernel.
> > >
> > > From the QuIP# paper, we know that doing INT4 quantization without the RHT (e.g. GPTQ 4 bit) results in significantly worse quality than QUIP#, which has worse quality than QTIP at all bitrates. This means we can’t actually use BitBLAS FP16 x INT4 to reach the same quality as 4 bit QTIP, *and* QTIP is faster. To reach the same quality as 4 bit QTIP, we would need INT8 quantization, which we already know is 1/3 slower than 4 bit QTIP. Finally, we note that compared to QuIP# and AQLM, our two main baselines, QTIP is both faster and higher quality at all bitrates. This means that QTIP strictly improves over those two VQ-based methods, showing that TCQ is indeed practical. We are also not professional kernel writers, so it is very possible that someone could write a faster kernel than us, giving further speedups.

---

> > > > ### Comment · Reviewer_ZTfk · 2024-08-09
> > > > **Extra Note**
> > > >
> > > > Thanks for the comparisons and extra explanations!
> > > >
> > > > Small note: There is nothing stopping the Nvidia hardware from doing mixed-precision kernels, e.g. Cutlass generates mixed-integer-precision like w4a8 kernels just fine :)

---

> > > > > ### Author Response · Authors · 2024-08-10
> > > > > **cutlass**
> > > > >
> > > > > NVIDIA TensorCores do not take mixed-precision inputs natively, so kernels like Cutlass and BitBLAS usually upcast the lower precision operand and do the matmul in the higher precision operand’s datatype. For memory bound inference, this doesn’t affect throughput since the decompression step is dominated by reading the weights and activations from memory.

---

> > ### Comment · Reviewer_1CRX · 2024-08-13
> >
> > Dear Authors,
> >
> > I have read the rebuttal. I will keep my positive rating.

---

> > > ### Author Response · Authors · 2024-08-13
> > > **Thanks**
> > >
> > > Thank you for your review!

---

### Official Review · Reviewer_3PHK · 2024-07-16

**Soundness:** 3
**Presentation:** 3
**Contribution:** 3
**Rating:** 6
**Confidence:** 3

**Summary:**

This work presents a new method of weight-only post-training quantization (PTQ) that uses trellis-coded quantization (TCQ) to achieve ultra-high-dimensional quantization. Although TCQ was introduced by Mao and Gray, QTIP transfers it into the LLM space and introduces new algorithms to make the method hardware efficient, thereby enabling fast inference.

**Strengths:**

1. The idea of using TCQ for LLM is pretty interesting and, to the best of my knowledge, new.
2. The paper is well-written, with explanatory figures and detailed explanations of the algorithms.
3. Paper includes a comprehensive amount of experiments.
4. The authors  provide extensive details to reproduce paper and promised to make the code public.

**Weaknesses:**

1. The ablation study is almost non-existent. For example, there is no analysis of the inference speed dependency on each introduced trick.Another thing you can do is experiment - perplexity (ppl) without the tricks, using L, k, and V values that still fit in the L1 cache, alongside the introduced algorithms.
2. The L, K, and V parameters were not explored. It would be beneficial to see different V values compared at the same bit width.
3. The paper contains a lot of notation and symbols that are either described once briefly or assumed to be known (e.g., see kd in line 27 line 128). This made reading the paper hard for me, as I had to constantly refer back to previous pages to understand what each symbol represents. For example, see line 229. This issue persists throughout the paper. Maybe it would be good to soft reintroduce symbols for time to time.
4. To be frank, the performance gains for me are not that impressive, except perhaps for LLama-3 at 2 bits.

**Questions:**

1. In the paper Line 236, you mentioned that QTIP can be fine-tuned. Did you try it? Did it provide any performance boost?
2. Can you please add the formula to calculate the average bit? While it can be deduced from the details given, having the formula would make readability clearer.
3. There are misspellings in line 46 and line 128: tk/kt , this should be corrected.
4. Is the formula in line 125 for the trellis structure correct? If we look at line 101, one of these cases seems to be a misspelling.
5. The numbers in Table 5 are not properly highlighted, especially where other methods are winning or matching.
6. The inference speed reported for AQLM is very odd. The AQLM paper claims around a 1.3x speed-up over FP16, but in your case, it is much slower. I suppose this may happen due to other library overheads like transformers, Pytorch and Python. Did you try the code with compiled CUDA graphs? Please take a look at [this example](https://colab.research.google.com/github/Vahe1994/AQLM/blob/main/notebooks/aqlm_cuda_graph.ipynb) from there repository.

Other:
1. Line 70 begins with "The effectiveness of these methods." It is a new section, and because of this, it is not clear which methods you are referring to.
2. Cite the linear congruential generator (LCG) paper.
3. In Line 240, should be mentioned that  AQLM doing block fine-tuning. Quip# doing full model fine-tuning. Full model fine-tuning also can be applied at AQLM, and the numbers will much Quip#. See Table 4 https://arxiv.org/pdf/2401.06118.

**Limitations:**

yes

---

> ### Author Rebuttal · Authors · 2024-08-07
>
> ## Ablations and fast inference (W1 and W2)
> The main components of QTIP that enable fast inference are the bitshift trellis and compute-based codes. To understand why these are necessary, let us look at the L, K, and V parameters in trellis coding. L is the trellis size, K is the bitrate, and V is how many weights we quantize per step. Larger trellises improve quality, and increasing V can increase decoding throughput by amortizing ops. However, a larger V enforces more structure on the search space and may decrease quality. When L = KV, TCQ becomes K bit V dim VQ, bridging the two.
>
> The bitshift trellis lets us avoid storing the trellis, which requires $2^{L+KV}L$ bits ($2^L$ nodes each with $2^{KV}$ edges). The bitshift trellis also enables parallel decoding, which is important since modern accelerators are highly parallel. The compute-based codes dissociate L from the amount of cache needed. While a pure-lookup codebook would take $16 \* 2^L V$ bits of cache, scaling exponentially with L, the compute-based codes use a constant amount or none at all. Note that this means we could have used L > 16 in the paper to achieve higher quality at the cost of *encoding* time, since L does not affect decoding speed.
>
> Table A shows an ablation on L for quantizing Llama 2 7B with K=2, V=1, the bitshift trellis, a pure-lookup codebook, and no fine-tuning. L=8 is the largest L achievable if we had to store the trellis *and* codebook in the same amount of cache as the HYB code (2KiB). L=10 is the largest L achievable if we only had to store the codebook. As expected, increasing L improves quality. Table A also shows very little difference between an equal-sized LUT codebook and QTIP’s codes, meaning that QTIP isn't sacrificing quality for speed. However, an equal-sized LUT would need >10X more cache than the latest GPUs have, making the bitshift trellis and compute-based codes necessary to achieve both quality and speed.
>
> Table B shows an ablation on V with L=12 and 16, K=2, and the same settings as Table A. Increasing V generally decreases quality, but this can be recovered with a larger L. It is hard to measure V's impact on decoding speed since this is highly implementation and hardware dependent, so V is more of a user-chosen hyperparameter.
> ### Table A
> |   L   | Trellis Size  |    CB size   |  total size  |  W2  |  C4  |
> |:-----:|:-------------:|:------------:|:------------:|:----:|:----:|
> | QuIP# |       -       |      8Kb     |      8Kb     | 8.22 | 11.0 |
> |   8   |    8.19 Kb    |    4.10 Kb   | **12.29 Kb** | 7.83 | 10.3 |
> |   10  |    40.96 Kb   | **16.38 Kb** |   57.34 Kb   | 7.49 | 9.67 |
> |   12  |   196.61 Kb   |   65.54 Kb   |   262.14 Kb  | 6.97 | 9.21 |
> |   16  |    4.19 Mb    |    1.05 Mb   |    5.24 Mb   | 6.83 | 8.92 |
> |   16  |    Bitshift   |     3INST    |      0Kb     | 6.82 | 8.96 |
>
> ### Table B
> |      Codebook     |  L | V |  W2  |  C4  |
> |:-----------------:|:--:|:-:|:----:|:----:|
> |        LUT        | 12 | 1 | 6.97 | 9.21 |
> |        LUT        | 12 | 2 | 7.09 | 9.24 |
> |        LUT        | 12 | 4 | 7.55 | 9.88 |
> |        LUT        | 16 | 1 | 6.83 | 8.92 |
> |        LUT        | 16 | 2 | 6.79 | 8.97 |
> | QTIP HYB (no FT)  | 16 | 2 | 6.83 | 8.97 |
> |        LUT        | 16 | 4 | 6.92 | 9.07 |
>
> ## Notation (W3)
> We will update the camera ready to improve readability.
>
> ## Performance Gains (W4)
> The easiest way to see QTIP’s improvements are at 4 bits and in larger models, where fine-tuning has less impact (Table 3). Tables 3 and 4 show that at 4 bits, all 3 of the QTIP formulations significantly reduce the quantization error over QuIP# and AQLM, which is impressive given how well these methods do at 4 bits. Table 3 also shows that for most model sizes and bitrates, QTIP *without fine tuning* matches or exceeds QuIP# *with fine tuning*, showing the importance of using a good quantizer. Finally, these quality gains are essentially “free” over QuIP#, since QTIP offers the same fast inference speeds.
>
> ## Fine-tuning QTIP (Q1)
> Yes, the HYB code has a small codebook that can be fine-tuned. Table 4 uses fine-tuning with the small codebook included in the set of tunable parameters. Tuning the codebook helps slightly, but a better fine-tuning algorithm could probably do better.
>
> ## Bit Accounting (Q2)
> QTIP has two sources of overhead beyond the quantized weights: sign vectors used for the random Hadamard transform (RHT), and the codebook in the HYB code (3INST and 1MAD do not have codebooks). For a nxm matrix, the sign vectors take up (n+m) bits. For the HYB code, the codebook takes up 2^Q x V x 16 bits (see Section 3.1.2 for more details). We used Q=9, but you can go down to Q~6 without noticeable degradation. At Q=9, the codebook uses 2KiB. Amortized over n*m entries, the codebook and sign vectors take up <0.01 additional bits. We will add these details to the appendix.
>
> ## tK vs Kt (Q3)
> tK and Kt = t times K
> ## L125 (Q4)
> 125 should say $2^L \times 2^{kV}$
> ## Highlighting (Q5)
> We will re-highlight the numbers to be more consistent.
>
> ## AQLM inference speed (Q6)
> The QuIP# authors recently released a more accurate generation timing script. We have re-timed all the methods on an RTX 6000 Ada. The numbers below are the average throughput for decoding 1024 tokens over 8 trials; we will update the paper with this.
>
> |    Method   |    2-7B    | 2-70B |
> |:-----------:|:----------:|:-----:|
> |     FP16    | 55.9 tok/s |  OOM  |
> |  AQLM 2 Bit |    81.5    |  8.78 |
> | QuIP# 2 Bit |     186    |  22.2 |
> |  QTIP 2 Bit |     188    |  23.5 |
>
> ## L70 (O1)
> “These methods” refers to those listed on L66-68. We will clarify this.
>
> ## LCG citation (O2)
> We will update the paper with this.
>
> ## AQLM fine-tuning (O3)
> Thank you for pointing this out, we will update the paper with these numbers. We note that this does not change our analysis, which is that QTIP achieves better performance than both QuIP# and AQLM with the same general fine-tuning scheme, while preserving QuIP#'s fast inference.

---

> > ### Comment · Reviewer_3PHK · 2024-08-12
> >
> > First, let me thank you for your response and for running the experiments. I have a few comments regarding your response:
> >
> > (W4) Generally, I would not recommend trusting PPL results at this scale. There is a risk of overfitting on the data. You can see this yourself by looking at Table 5 at zero-shot results. While QTIP ppl constantly better at 4 bits (although by not much), the difference at zero shots are not significant.
> >
> > (Q1) This was not clear to me from reading the paper. I would suggest making it clearer wherever you are using fine-tuning version.
> >
> > (W1, W2) I would suggest adding this information in the appendix, if it fits.
> >
> > (Q6, W3) Great, thank you!
> >
> > I will keep my score.

---

> > > ### Author Response · Authors · 2024-08-13
> > > **Thanks**
> > >
> > > Thank you for your review, we will be sure to add clarifying information into an updated manuscript.

---

### Author Rebuttal · Authors · 2024-08-07

We thank the reviewers for their detailed reviews. As multiple reviewers noted, QTIP is well motivated and novel (1CRX, 3PHK, ZTfk), and simultaneously achieves strong quality and fast inference (1CRX, F569, ZTfk). Below, we have written individual responses to reviewers. We have also run a number of new experiments. These include ablations on L and V (3PHK), more accurate timing numbers with an updated script (3PHK), 3 and 4 bit inference kernels (1CRX), and timing on different GPUs (1CRX).

**To satisfy the character count, we have abbreviated the weaknesses/questions we are responding to. "WX" referrs to weakness X and "QY" question Y according to the order in the review.**

---

### Decision · Program_Chairs · 2024-09-25

**Decision:**

Accept (spotlight)

**Comment:**

Application of TCQ for quantizing trained weights results in strong performance and fast inference, as the reviewers mostly agreed. TCQ is a well-established practice in the communicaiton field and this paper is mainly about borrowing it to model qualitization in ML; neverthless, bitshift trellis and compute-based random Gaussian codes are important new features and the value is significant. Ablation requests were complied to the satsifaction of the reviewer. Studies on larger bit width and variety of devices were also added, which made the work more convincing.